# Targeted Sequential Indirect Experiment Design

**Elisabeth Ailer**
Technical University of Munich
Helmholtz Munich
Munich Center for Machine Learning (MCML)

**Niclas Dern**
Technical University of Munich

**Jason Hartford**
Valence Labs

**Niki Kilbertus**
Technical University of Munich
Helmholtz Munich
Munich Center for Machine Learning (MCML)

## Abstract

Scientific hypotheses typically concern specific aspects of complex, imperfectly understood or entirely unknown mechanisms, such as the effect of gene expression levels on phenotypes or how microbial communities influence environmental health. Such queries are inherently causal (rather than purely associational), but in many settings, experiments can not be conducted directly on the target variables of interest, but are indirect. Therefore, they perturb the target variable, but do not remove potential confounding factors. If, additionally, the resulting experimental measurements are multi-dimensional and the studied mechanisms nonlinear, the query of interest is generally not identified. We develop an adaptive strategy to design indirect experiments that optimally inform a targeted query about the ground truth mechanism in terms of sequentially narrowing the gap between an upper and lower bound on the query. While the general formulation consists of a bi-level optimization procedure, we derive an efficiently estimable analytical kernel-based estimator of the bounds for the causal effect, a query of key interest, and demonstrate the efficacy of our approach in confounded, multivariate, nonlinear synthetic settings.

## 1 Introduction

Experimentation is the ultimate arbiter of scientific discovery. While advances in machine learning (ML) and ever increasing amounts of observational data promise to accelerate scientific discovery in virtually all scientific disciplines, all hypotheses ultimately have to be supported or falsified by experiments. But the space of possible experiments is combinatorial, and as a result, experimentation in the physical world may become a major bottleneck of the scientific process and critically inhibit, for example, fast turnaround in drug discovery loops even in fully automated labs. Efficient adaptive data-driven strategies to propose the most useful experiments are thus of vital importance.

Targeted experimentation requires a well-formed hypothesis about the underlying system. We focus on hypotheses that are typically expressed in terms of causal effects or properties of functional relationships. For example, we may posit that there is some ground truth mechanism/function $f$ that describes how a given phenotype depends on gene expression levels in a cell, or how the composition of microbes affects certain environmental health indicators. In the natural sciences, such mechanisms are typically (a) nonlinear, (b) dependent on multi-variate inputs, and (c) confounded by additional, unobserved quantities. For example, the way in which the gut microbiome composition affects energy metabolism in humans is likely confounded by various environmental and lifestyle choices. Due to

38th Conference on Neural Information Processing Systems (NeurIPS 2024).

these factors, $f$ is generally unidentifiable, i.e., we cannot hope to infer $f$ from (even infinite amounts of) observational data alone.

Scientific queries are typically more narrow: 'Can I expect changes in body mass index when I increase the relative abundance of Lactobacillus bacteria in the gut? How does the susceptibility of breast cancer depend on BRCA1 expression levels?' Such targeted scientific queries do not require full knowledge of the underlying mechanism $f$, but only concern a specific aspect of $f$, which we can express mathematically as a functional $Q$ of $f$. Due to potential unobserved confounding, even targeted queries $Q[f]$ may not be identified from observational data alone (Bennett et al., 2022)—experimentation is required. In these examples, the inputs to $f$ of interest – which we call *treatments* – are multi-variate measurements such as gene expression levels or microbiome abundances. Randomizing these treatment variables is typically thought of as the gold standard for removing confounding effects, but perfect randomization is typically impossible: we can seldom perform hard interventions that set a distribution over these variables' outcomes independently of confounding factors. Instead, experimental access in the natural sciences often amounts to limited possible perturbations of the system that induce distribution shifts of the treatments. For example, while we can administer antibiotics with very predictable effects on certain microbes, such an intervention does not remove confounding effects from lifestyle choices such as nutrition and exercise or environmental factors. Therefore, we think of experimentation as a source of indirect interventions, which strongly perturb the distribution over the treatment variables of interest.

Our primary goal is to design experiments that maximally inform the query of interest, $Q[f]$, within a fixed budget of experimentation. For multi-variate treatments and nonlinear $f$ the query $Q[f]$ may not be identifiable at all, or only be identified after an infeasible number of experiments. We address this by maintaining an upper and lower bound on the query, and sequentially selecting experiments that minimize the gap between these bounds. We show that by treating experiments as instrumental variables (IV) we can estimate the bounds by building on existing techniques in (underspecified) nonlinear IV estimation. Our procedure involves a bi-level optimization, where an inner optimization estimates the bounds, and an outer optimization seeks to minimize the gap between the bounds. We show that if we are prepared to assume that $f$ lies within a reproducing kernel Hilbert space (RKHS), then there are queries $Q[f]$ for which we can solve the inner estimation in closed form. In summary, we make the following contributions:

- We formalize the problem of indirect adaptive experiments in nonlinear, multi-variate, confounded settings as a sequential underspecified instrumental variable estimation, where we seek to adaptively tighten estimated bounds on the target query.
- We derive closed form expressions for the bound estimates when assuming $f$ to lie within a reproducing kernel Hilbert space (RKHS) for linear $Q$.
- We develop adaptive strategies for the outer optimization to iteratively tighten these bounds and demonstrate empirically that our method robustly selects informative experiments leading to identification of $Q[f]$ (collapsing bounds) when it can be identified via allowed experimentation.

## 2 Problem Setting and Related Work

### 2.1 Problem Setting

**Data generating process.** We assume the following data generating process

$$X = h(Z, U), \qquad Y = f_0(X) + U, \tag{1}$$

where we have experimental access to $Z \in \mathcal{Z} \subset \mathbb{R}^{d_z}$, $X \in \mathcal{X} \subset \mathbb{R}^{d_x}$ are the actual inputs to the mechanism of interest, i.e., the 'treatments' (in the cause-effect estimation sense) or 'targets' (in the indirect experimentation sense), $Y \in \mathcal{Y} \subset \mathbb{R}$ is the scalar outcome, e.g., the phenotype of interest, and $U$ is an unobserved confounding variable with $\mathbb{E}[U] = 0$. The function $h$ determines how experiments (choices of $Z$) affect treatments $X$ and can be arbitrary. The function $f_0$ is the mechanism of interest for our scientific query and can also be arbitrary. As in typical indirect experiment settings, we assume that $Z$ consists of valid instrumental variables: (1) $Z \not\!\perp\!\!\!\perp X$, i.e., we do perturb the distribution of $X$ by experimenting on $Z$, (2) $Z \perp\!\!\!\perp U$, i.e., our choice of experiments is not influenced by the confounding variables, and (3) $Z \perp\!\!\!\perp Y \mid X, U$, i.e., $Z$ only affects $Y$ via $X$ and does not have a direct effect on $Y$. When $X$ is multi-variate and $f_0, h$ are allowed to be non-linear, $f_0$ is commonly not

identified form observational data. However, we argue that in such settings one should be aiming for a more targeted query, i.e., does not aim to identify $f_0$ fully, but only a certain aspect of the mechanism.

**Scientific queries.** We represent the scientific query of interest by a functional $Q : \mathcal{F} \to \mathbb{R}$ where $\mathcal{F} \subset L_2(X)$ is the space of considered mechanisms $f$.[1] Generic functionals can measure both local as well as global properties of any $f \in \mathcal{F}$: simple average treatment effects via $Q[f] := \mathbb{E}[Y|do(X = x^*)] = f(x^*)$; when $\mathcal{F}$ is a Hilbert space, projections of $f$ onto a fixed basis function $f^*$ via $Q[f] := \langle f^*, f \rangle_{\mathcal{F}}/\|f\|_{\mathcal{F}}^2$; the local causal effect of an individual component $X_i$ on $Y$ at $x^*$ via $Q[f] := (\partial_i f)(x^*)$.[2] Here, $x^*$ could be the mean of the observed treatments, i.e., represent a 'base gene expression level' and we are interested in how a local change away from that base level affects the outcome. While our methodology applies to any functional, we focus our empirical experiments on causal effects of individual components as key scientific queries, such as how does up- or down-regulating a given gene affect the phenotype?

**Learning experiments.** Our goal is to sequentially learn a policy $\pi \in \mathcal{P}(\mathcal{Z})$[3] such that data sampled from the joint distribution $P_\pi(X, Y, Z) = \pi(Z)P(X \mid Z)P(Y \mid X)$ induced by the model in eq. (1) optimally informs $Q[f_0]$. We highlight that depending on $f_0, h$ and the distribution of $U$, there may exist a policy $\pi$ such that $Q[f_0]$ is identified from $P_\pi(X, Y, Z)$, but in general it may remain unidentified for all policies even in the infinite data limit. For a non-optimal policy, i.e., non-informative experimentation, we should expect $Q[f_0]$ to be partially identified from $P(X, Y, Z)$ at best. Therefore, we aim to estimate upper and lower bounds $Q^+(\pi), Q^-(\pi)$ of $Q[f_0]$ and sequentially learn the policy $\pi$ that minimizes $\Delta(\pi) := Q^+(\pi) - Q^-(\pi)$. In each round $t \in [T] := \{1, \ldots, T\}$, we observe $n$ i.i.d. samples from $P_{\pi_t}(X, Y, Z)$ using policy $\pi_t$, which we use (potentially together with data collected under previous policies $\pi_{<t}$) to estimate $Q^+(\pi_t), Q^-(\pi_t)$ and then aim to propose a new policy $\pi_{t+1}$ that yields a smaller gap in bounds: $\Delta(\pi_{t+1}) < \Delta(\pi_t)$.

**Connections to other settings.** Besides the 'indirect experimentation' lens, the setting in eq. (1) can be viewed from different perspectives. While they all share the same mathematical formulation, they start from different conceptions of $Z$. The literature on *invariant causal learning* (Peters et al., 2016; Heinze-Deml et al., 2018) may interpret $Z$ in eq. (1) as an environment indicator and $f_0$ is invariant across these environments. Each policy corresponds to an environment ($Z$ cannot be designed, but we can collect data for different $Z$) with its own distribution of $X$. The heterogeneity across environments can be leveraged to learn about the invariant mechanism $f$. In *instrumental variables* (Pearl, 2009; Angrist & Pischke, 2008), one typically assumes to encounter a situation as in eq. (1) with the assumptions (1)-(3) where $X, Y, Z$ are observed 'in the wild'. Valid instruments $Z$ then sometimes allow for (partial) identification and estimation of $f_0$ despite the unobserved confounder $U$. Again, there is typically no active experimentation in IV estimation, but $Z$ is taken 'as is'. Finally, experimentation on $Z$ in our setting can also be interpreted as *soft interventions* (Jaber et al., 2020; Pearl, 2009) on $X$, where we intervene on $X$ setting it to follow a given distribution instead of a fixed value. Instead of allowing arbitrary soft interventions, eq. (1) restricts us to distributions $P(X \mid Z)\pi(Z)$ for feasible experimental policies $\pi(Z)$. Importantly, unlike proper soft interventions, we do not get rid of unobserved confounding. Ultimately, we face a sequence of IV estimation tasks. Experimental access to $Z$ justifies the instrumental variable assumption (2) $Z \perp\!\!\!\perp U$, and indirect experiments are setup by design such that (1) $Z \not\perp\!\!\!\perp X$ holds. Assumption (3) $Z \perp\!\!\!\perp Y \mid X, U$ instead may be hard to justify in practice and limits the types of experimentation allowed in our framework.

## 2.2 Related Work

**IV estimation.** Instrumental variables are a cornerstone of cause-effect estimation in econometrics (Angrist & Pischke, 2008) at least since their use by Philip G. Wright in 1928 (Wright, 1928; Stock & Trebbi, 2003). Even though valid instruments are hard to come by in practice (Hernán & Robins, 2006), generalizations to settings with relaxed assumptions have sparked renewed interest in IV estimation from the machine learning community not least due to successful applications in Mendelian randomization (Sanderson et al., 2022; Didelez & Sheehan, 2007; Legault et al., 2024) or on microbiome data (Sohn & Li, 2019; Wang et al., 2020; Ailer et al., 2021). In particular, we build on recent advances in nonlinear/nonparametric IV estimation (Newey & Powell, 2003; Lewis & Syrgkanis, 2018; Singh et al., 2019; Hartford et al., 2017; Saengkyongam et al., 2022; Zhang

---

[1] $L_2(X)$ is the $L_2$-space of scalar-valued functions of $X$ with respect to the distribution of $X$.

[2] We write $(\partial_i f)(x^*)$ for the partial derivative of $f$ with respect to the $i$-th argument evaluated at $x^*$.

[3] Here, $\mathcal{P}(\mathbb{R}^{d_z})$ denotes the space of probability distributions over $\mathbb{R}^{d_z}$.

et al., 2020; Xu et al., 2020), with a specific focus on the minimax formulation using the generalized method of moments (Liao et al., 2020; Dikkala et al., 2020; Bennett et al., 2023; Zhang et al., 2023; Liao et al., 2020; Bennett et al., 2019; Muandet et al., 2019; Bennett et al., 2022). In addition, we do not assume identifiability in line with recent approaches to partial identification and bounding causal effects in IV settings relaxing various discreteness and additivity assumptions (Kilbertus et al., 2020; Padh et al., 2022; Frauen et al., 2024; Melnychuk et al., 2024; Wang & Tchetgen Tchetgen, 2018; Gunsilius, 2019; Hu et al., 2021; Zhang & Bareinboim, 2021).

**Sequential experiment design.** The work closest to ours is perhaps by Ailer et al. (2023), where they consider a similar setting of sequential indirect experiment design via IV estimation. The key differences are that they only consider a fully linear setting ($h$, $f_0$ linear), aim for full identification of $f_0$ (even though the setting is underspecified), and only provide non-adaptive strategies, i.e., the sequence of experiments does not depend on data collected throughout the experiments. Other work on indirect experimentation either assumes no unobserved confounding (Singh, 2023) or focuses primarily on sample efficiency and variance reduction when aiming for full identification of $f$ (Chandak et al., 2024). Adaptive learning has also been used in another context for cause effect estimation by 'deciding what to observe' Gupta et al. (2021). While clearly related, we highlight that most of the literature on active experimentation for causality aims at learning the causal structure (instead of estimating properties of $f$) from access to interventions, i.e., direct experiments (instead of more realistic indirect experiments), e.g., (von Kügelgen et al., 2019; Sverchkov & Craven, 2017; Agrawal et al., 2019; He & Geng, 2008; Gamella & Heinze-Deml, 2020; Elahi et al., 2024; Zemplenyi & Miller, 2021). Additionally, we require a strategy that is path-dependent, while in active learning the objective function remains the same over the different iterations.

## 3 Methodology

### 3.1 Minimax Instrumental Variable Estimation

For the general instrumental variable setting in eq. (1) that we face in each round, we aim to solve $\mathbb{E}[Y - f(X) \mid Z] = 0$. To solve this, we follow the conditional moment formulation (Bennett et al., 2019; Dikkala et al., 2020; Bennett et al., 2022, 2023), which builds on the observation that we only want to consider functions $f$, such that the residuals, $U_f := Y - f(X)$ are independent of the instrument $Z$. If two random variables are independent, $U_f \perp\!\!\!\perp Z$, then $E[g(Z)U_f] = E[g(Z)]E[U_f]$ for all test functions $g$. We search for $f$ that minimize the residual while maintaining the independence constraint by solving a minimax optimization that minimizes $f$ over a worst case $g$. The integral equation $\mathbb{E}[Y - f(X) \mid Z] = 0$ for $f$ can be written as $Tf = r_0$, where $T$ is a linear, bounded operator mapping $f : \mathcal{X} \to \mathbb{R}$ to $Tf := \mathbb{E}[f(X) \mid Z] : \mathcal{Z} \to \mathbb{R}$ and $r_0 = \mathbb{E}[Y \mid Z]$. This is typically an ill-posed inverse problem where both $T$ and $r_0$ are not known but have to be estimated from i.i.d. data $\mathcal{D} = \{(x_i, y_i, z_i)\}_{i=1}^n$. Assuming $f$ to come from a set of hypothesis functions $\mathcal{F} \subset L_2(X)$ and $\mathcal{G} \subset L_2(Z)$ a class of 'witness functions' or 'test functions', we can write the non-parametric IV problem as a minimax optimization problem of the form (Dikkala et al., 2020; Bennett et al., 2023)

$$\arg\inf_{f \in \mathcal{F}} \sup_{g \in \mathcal{G}} L(f, g) \tag{2}$$

for some objective function $L$ mapping tuples of hypotheses $f$ and test functions $g$ to $\mathbb{R}$. While most existing approaches start by assuming that there exists a unique solution of $Tf = r_0$ (Lewis & Syrgkanis, 2018; Liao et al., 2020; Muandet et al., 2019; Dikkala et al., 2020; Bennett et al., 2019), based on Lagrange multipliers Bennett et al. (2023) recently have shown that under a source condition on $T$ and mild realizability assumptions regarding the capacity of the (statistically restricted) function classes $\mathcal{F}$ and $\mathcal{G}$ (Bennett et al., 2023, Assumptions 2, 3, 4), the solution of $Tf = r_0$ is given by

$$f_0 = \arg\min_{f \in \mathcal{F}} \max_{g \in \mathcal{G}} \tfrac{1}{2}\|f\|_{L_2(X)}^2 + \langle r_0 - Tf, g \rangle_{L_2(Z)} = \arg\min_{f \in \mathcal{F}} \max_{g \in \mathcal{G}} \tfrac{1}{2}\mathbb{E}[f^2(X)] + \mathbb{E}[(Y - f(X))g(Z)].$$

$$\tag{3}$$

This formulation of the objective is immediately amenable to empirical estimation from $\mathcal{D}$ by using empirical averages for the expectations. Eq. (3) can be viewed as a penalized optimization, where $\tfrac{1}{2}\mathbb{E}[f^2(X)]$ penalizes the solution towards the minimum $L_2$ norm.

The inner maximization over test functions ensures that we only consider hypotheses $f \in \mathcal{F}$ compatible with the conditional moment restrictions that enforce the required independence of the residuals. While there may still be multiple hypotheses $f \in \mathcal{F}$ compatible with observational data in

this way, Bennett et al. pick the $f$ with the least $L_2$ norm, yielding a unique solution. In the following, we build on this intuition to estimate bounds on a chosen scientific query $Q$ among all $f$ compatible with the observed data and assumed structure (via the moment restrictions).

## 3.2 Sequential Policy Learning for Bounding Functionals

The source condition required for eq. (3) is that $r_0$ lies in the range of $TT^*$, where $T^*$ is the adjoint of $T$. As Bennett et al. point out, this is a nontrivial restriction and paramount in ensuring uniqueness in eq. (3). Without this source condition the IV problem is still ill-posed in that there may be multiple solutions $f \in \mathcal{F}$ compatible with the observed data $\mathcal{D}$ (even in the infinite data limit). One of our key realizations is that we can still meaningfully make use of the penalized optimization formulation in eq. (3) to compute bounds on targeted queries about $f$.

Let $Q : \mathcal{F} \to \mathbb{R}$ be a functional on hypotheses $\mathcal{F}$ that captures the scientific query. For any given joint distribution $P_\pi(X, Y, Z)$ induced by experimental policy $\pi$, to bound $Q[f_0]$, we can compute

$$Q^{\pm}(\pi) = \inf_{f \in \mathcal{F}} \sup_{g \in \mathcal{G}} \pm Q[f] + \mathbb{E}[(Y - f(X))g(Z)], \qquad (4)$$

where again the inner maximization aims at only considering hypotheses compatible with the conditional moment restrictions $\mathcal{F}_c \subset \mathcal{F}$ and the outer minimization aims at finding the $f \in \mathcal{F}_c$ with the largest/smallest value of $Q[f]$. The realizability assumption that $f_0 \in \mathcal{F}$ then clearly implies that $f_0 \in \mathcal{F}_c$ and the bounds $Q^{\pm}$ will contain $Q[f_0]$. We note that specially when $X$ is of higher dimension than $Z$, in practice neither $f$ nor the much more narrow query $Q[f]$ are guaranteed to be identified. However, depending on $\pi$, we may shape $P_\pi$ such that $\mathcal{F}_c$ becomes restricted enough to obtain informative bounds $Q^{\pm}$ on $Q[f_0]$. Bennett et al. (2022) work out in detail when a linear functional of $f_0$, but not $f_0$ itself, is identified. If $P_\pi$ is such that $T$ satisfies the source condition, certain queries $Q$ (such as the $Q[f] = \frac{1}{2}\|f\|_{L_2}^2$ as a canonical example) may actually be identified and the bounds $Q^{\pm}$ will coincide.

We can now formulate our main policy learning goal as

$$\inf_{\pi \in \Pi(\mathcal{Z})} \Delta(\pi), \qquad \text{where } \Delta(\pi) = Q^+(\pi) - Q^-(\pi), \qquad (5)$$

with $\Pi(\mathcal{Z}) \subset \mathcal{P}(\mathcal{Z})$ being a subset of implementable experimentation policies. In practice, we aim at approximating this optimization via sequential updates of $\pi_t$ over $T$ rounds, where we observe $n$ i.i.d. samples of $P_{\pi_t}$ at each round. This means that given $\pi_t$, we seek to choose $\pi_{t+1}$ such that $\Delta(\pi_{t+1}) < \Delta(\pi_t)$. The final output of our method is the interval $[Q^-(\pi_T), Q^+(\pi_T)] \ni Q[f_0]$. A key advantage of this formulation is that we need not assume identifiability of $f$ nor of $Q[f]$, will obtain valid (albeit potentially loose) bounds when the experimentation budget $T$ is restricted, and that potential identifiability will be 'captured automatically' by the bounds coinciding. If after $T$ rounds the bounds are still not informative, the experimenter can choose more expressive experiments (different $\Pi(\mathcal{Z})$ and even different $\mathcal{Z}$), a more specific query $Q$, or put stronger assumptions on the hypothesis class $\mathcal{F}$.

## 3.3 Solving the Minimax Optimization Problem

Let us unpack the sequential policy learning problem. At each round $t \in [T]$, we need to solve two minimax problems for $Q^{\pm}(\pi_t)$ where the objective is estimated from finite data and then take a minimization step over the difference of the two solutions. To further complicate this 'tri-level' (min + minimax) optimization problem, we are optimizing over two function spaces $(\mathcal{F}, \mathcal{G})$ and a family of distributions $(\Pi(\mathcal{Z}))$. In the remainder of this section we develop techniques to render these optimizations feasible in practice for suitable choices of $\mathcal{F}, \mathcal{G}, \Pi(\mathcal{Z})$.

Existing solutions for the minimax problems typically employ adversarial optimization (iterative gradient-based optimization over parametric function spaces) or kernel-based techniques. Since the former are usually hard and expensive to optimize reliably even without the outer minimization, we focus on the latter approach using reproducing kernel Hilbert spaces (RKHS) for test functions $\mathcal{G} = \mathcal{H}_Z$ induced by a characteristic kernel $k_Z : \mathcal{Z} \times \mathcal{Z} \to \mathbb{R}$. Then there exists a closed form expression for a consistent estimate of the inner supremum.

**Theorem 1** (Zhang et al. (2020)). *If* $\mathbb{E}[(Y - f(X))^2 k_Z(Z, Z)] < \infty$,

$$\left( \frac{1}{n^2} \sum_{i=1}^{n} \sum_{j=1}^{n} (y_i - f(x_i)) k(z_i, z_j)(y_j - f(x_j)) \right)^{\frac{1}{2}} \tag{6}$$

*consistently estimates* $\sup_{g \in \mathcal{H}_Z, \|g\| \leq 1} \langle r_0 - Tf, g \rangle_{\mathcal{H}_Z}$ *from data* $\mathcal{D}$.

Hence, for $n$ samples $\mathcal{D} \overset{\text{iid}}{\sim} P_\pi$ we reduce the minimax optimizations in eq. (4) to

$$Q^{\pm}(\pi) = \inf_{f \in \mathcal{F}} \mp Q[f] + \left( \frac{1}{n^2} \sum_{i=1}^{n} \sum_{j=1}^{n} (y_i - f(x_i)) k(z_i, z_j)(y_j - f(x_j)) \right)^{\frac{1}{2}} . \tag{7}$$

One option to solve eq. (7) efficiently is via gradient-based methods leveraging highly efficient auto-differentiation packages for parametric function families, such as choosing neural networks for $\mathcal{F}$, i.e., we optimize over a finite-dimensional real-vector $\theta$—the weights of the neural network. Our framework may also benefit from other existing approaches to the minimax optimization from the literature, e.g., in Muandet et al. (2019); Dikkala et al. (2020). Parametric functions may have the advantage that they render the computation of $Q[f_\theta]$ feasible. For example, the causal effect of $X_i$ on $Y$ at $x^*$ given by $Q[f_\theta] = (\partial_i f_\theta)(x^*)$ can be obtained directly from automatic differentiation as well. Global functionals such as $Q[f_\theta] = \int_{\mathcal{X}} \psi(x) f_\theta(x)\, dx$ for some function $\psi$ may pose greater difficulties as the integral over $\mathcal{X}$ is typically infeasible. Still, this approach ultimately requires a bi-level optimization with a substantial amount of tuning.

In order to also obtain a closed-form estimate of the minimization over $\mathcal{F}$, we have to specify the functional $Q$. To this end, we focus on bounded linear functionals. One example is the causal-effect of an individual input $(\partial_i f)(x^*)$, because it is arguably one of the most common and relevant scientific queries: how does a small change of a specific variable affect the outcome? When following a fully non-parametric approach by also assuming hypotheses to come from an RKHS $\mathcal{F} = \mathcal{H}_X$ induced by a characteristic kernel $k_X : \mathcal{X} \times \mathcal{X} \to \mathbb{R}$, we can also estimate the minimization over $\mathcal{F}$ in closed form for the causal effect—a local, linear functional. The key realization is that functionals of functions represented as linear combinations of RKHS functions can be written as linear functions of the coefficients.

**Theorem 2.** *Assume functions in* $\mathcal{H}_Z, \mathcal{H}_X$ *to have bounded variation, (w.l.o.g.) images contained in* $[-1, 1]$, *and for* $h \in \mathcal{H}_Z$ *also* $-h \in \mathcal{H}_Z$. *Denote by* $K_{XX}, K_{ZZ} \in \mathbb{R}^{n \times n}$ *the empirical kernel matrices of* $\mathcal{D}$, *let* $k_X$ *be continuously differentiable, and let* $\lambda_g, \lambda_f, \lambda_c \geq 0$. *Further, fix* $Q[f]$ *to be a bounded linear functional, and write* $f_\theta(\cdot) = \sum_{i=1}^{n} \theta_i k(x_i, \cdot)$. *Then the solutions to the regularized minimax problems*

$$f^{\pm} = \arg\min_{f \in \mathcal{H}_X} \mp Q[f] + \lambda_c \sup_{g \in \mathcal{H}_Z} \langle r_0 - Tf, g \rangle_{\mathcal{H}_Z} + \lambda_f \|f\|_{\mathcal{H}_X} - \lambda_g \|g\|_{\mathcal{H}_Z} \tag{8}$$

*are consistently estimated by* $f_{\hat{\theta}^{\pm}}$ *with*

$$\hat{\theta}^{\pm} = (K_{XX} K_{ZZ} K_{XX} + 4 \frac{\lambda_g \lambda_f}{\lambda_c} K_{XX})^+ \left( K_{XX} K_{ZZ} y \mp \frac{1}{\lambda_c} Q[K_{X\cdot}](x^*) \right) , \tag{9}$$

*where* $(Q[K_{X\cdot}])(x^*) \in \mathbb{R}^n$ *is the vector* $((Q[k_X(x_1, \cdot)])(x^*), \ldots, (Q[k_X(x_n, \cdot)])(x^*)$. *Note that these general bounded linear functionals can be computed analytically for many common kernels such as linear, polynomial, or radial basis function (RBF) kernels.*

One common example is the partial derivative at $x^*$:

$$\hat{\theta}^{\pm} = (K_{XX} K_{ZZ} K_{XX} + 4 \frac{\lambda_g \lambda_f}{\lambda_c} K_{XX})^+ \left( K_{XX} K_{ZZ} y \mp \frac{1}{\lambda_c} (\partial_i K_{X\cdot})(x^*) \right) , \tag{10}$$

where $(\partial_i K_{X\cdot})(x^*) \in \mathbb{R}^n$ is the vector $((\partial_i k_X(x_1, \cdot))(x^*), \ldots, (\partial_i k_X(x_n, \cdot))(x^*)$.

We defer all proofs of our theoretical statements to Appendix A.

We can now substitute these closed-form estimates for the minimax problems into our policy learning objective $\Delta(\pi) = Q[f_{\hat{\theta}^+}] - Q[f_{\hat{\theta}^-}]$ in eq. (5)

$$\Delta(\pi) = -\frac{2}{\lambda_c} \left( (K_{XX} K_{ZZ} K_{XX} + 4 \frac{\lambda_g \lambda_f}{\lambda_c} K_{XX})^+ (Q[K_{X\cdot}])(x^*) \right)^{\top} (Q[K_{X\cdot}])(x^*) . \tag{11}$$

**(In)validity of bounds.** Our general policy learning formulation in eqs. (4) and (5) involves nonlinear, nonconvex optimizations over function spaces and families of distributions. Since we cannot provably obtain global optima to those, we cannot guarantee valid bounds. Even for the RKHS setting in Theorem 2 for linear $Q$, while the estimates $\hat{\theta}^{\pm}$ are consistent, for finite sample estimation we have to regularize both the hypotheses and witness functions ($\lambda_g, \lambda_f > 0$) to obtain numerically stable and reliable estimates. Clearly, those pose restrictions on the search spaces for $f, g$ and may thus lead to invalid bounds. We may expect real-world mechanisms $f_0$ to be relatively smooth such that these regularization terms do not render our estimates invalid in practice. Moreover, since we made no explicit assumptions about the functional $Q$, using $Q[f]$ as a penalty (as compared to, e.g., $\frac{1}{2}\|f\|_{L_2}$) may not yield a unique solution to eq. (8) even in the infinite data limit and for $\lambda_g = \lambda_f = 0$. While uniqueness can be recovered for strictly convex, coercive, and lower semicontinuous $Q$, this implies non-trivial restrictions on the allowed scientific queries, which we may not be willing to tolerate. Therefore, we introduce a 'penalization parameter' $\lambda_c$ (in front of the supremum term rather than $Q[f]$ for convenience) that allows us to empirically trade off the scale/importance of $Q[f]$ and the conditional moment restriction in practice. Enforcing conditional moment restrictions via a minimax formulation from finite samples typically exhibits high variance. Therefore, a lack of provably valid bounds due to practical regularization does not impair the usefulness of our method in a setting where we aim to learn about a potentially unidentified scientific query from limited experimental access. In particular, we argue that overly conservative bounds are still more useful than likely invalid point estimates based on one-size-fits-all assumptions required for identifiability.

Continuing with the closed-form expression in eq. (11) for the gap between maximally and minimally possible values of the scientific query – here the causal effect of changing one treatment component around a base level $X^*$ – over all hypotheses that are compatible with the observed data, we can now seek to represent the experimentation strategy $\pi$ in a way that allows us to sequentially improve it.

### 3.4 Sequential Experiment Selection

We now describe different approaches to sequentially update the experimentation policy. The experimentation budget $T$ and the number of samples per round $n$ are fixed. We denote by $\pi_t$ the policy in round $t$ and by $\mathcal{D}^{(t)}$ ($\mathcal{D}^{(\leq t)}$) the data collected in (up to) round $t$. We denote by $Q_t^{\pm}$ and $\Delta_t$ the bounds and gap between them, respectively, estimated in round $t$ and explicitly mention which data was used for the estimates. Some strategies rely on a 'meta-distribution' over policies $\Pi(\mathcal{Z})$, which we denote by $\Gamma$. A sample $\pi \sim \Gamma$ is thus a policy in $\Pi(\mathcal{Z}) \subset \mathcal{P}(Z)$. While the 'final result' of all strategies are just the final bounds $Q_T^{\pm}$, we report $Q_t^{\pm}$ also for suitable $t < T$ for different strategies to compare efficiency.

**Simple baseline.** In modern methods for experiment design, random exploration is often surprisingly competitive (Ailer et al., 2023). We consider a simple entirely non-adaptive baseline called **random**, which independently samples $\pi_t \sim \Gamma$ for $t \in [T]$ and estimates $Q_t^{\pm}$ on $\mathcal{D}^{(\leq t)}$.

**Locally guided strategies.** Next, we look to simple adaptive strategies that leverage the locality of $Q$. If $Q$ is determined in a small neighborhood around some $x^* \in \mathcal{X}$, a useful guiding principle for $\pi$ is to aim at concentrating the mass of the marginal $P_\pi(X)$ around $x^*$. The causal effect $(\partial_i f)(x^*)$ or the average treatment effect $\mathbb{E}[Y \mid do(X = x^*)]$ are examples of local queries. Estimating such local relationships in indirect experimentation without unobserved confounding has recently been studied from a theoretical perspective by Singh (2023). They propose a simple explore-then-exploit strategy and prove favorable minimax convergence rates depending on the complexities of $h$ and $f_0$ as well as the noise levels in the unconfounded setting, i.e., $X$ and $Y$ have independent noises and $h$ does not depend on $U$. We highlight that these strategies do not use the intermediate estimated bounds as feedback to select the next policy $\pi$. Pseudocode for the different strategies is in Appendix G.

1. **Explore then exploit (EE)** (inspired by Singh (2023)): Split the budget into $T = T_1 + T_2$. Sample $\pi_t \sim \Gamma$ for $t \in [T_1]$. Then, fit $\pi$ as a (parametric) distribution on the $Z$ values of the $K \leq T_1 \cdot n$ samples in $\mathcal{D}^{(\leq T_1)}$ closest to $x^*$ (i.e., their $X$ values have the smallest distance to $x^*$ for some suitable distance measure on $\mathcal{X}$). Set $\pi_t := \pi$ for $t \in \{T_1 + 1, \ldots, T\}$ and estimate $Q_t^{\pm}$ from $\bigcup_{t=T_1+1}^{T} \mathcal{D}^{(t)}$. The split between $T_1$ and $T_2$ may be informed by $n$ to maximize exploration while retaining sufficiently many samples during exploitation for a low variance estimate of $Q_T^{\pm}$.

2. **Alternating explore exploit (AEE)**: Throughout all rounds, keep a list of all observed samples sorted by the distance of their $X$ values to $x^*$. For odd $t \in [T] \setminus \{T\}$ independently sample

$\pi_t \sim \Gamma$. For even $t \in [T]$ and $t = T$ fit $\pi_t$ as a (parametric) distribution to the $\min\{K, n \cdot t\}$ nearest samples to $x^*$ for some $K \in \mathbb{N}$ and estimate $Q_t^{\pm}$ on $\bigcup_{\text{even } t} \mathcal{D}^{(t)}$.

There is some freedom in which data is used to estimate $Q_t^{\pm}$ and using $\mathcal{D}^{(\leq t)}$ is always a valid choice. For local queries it can still be beneficial in practice to discard data far away from $x^*$. More broadly, the above strategies could potentially be further improved by weighing samples in $\mathcal{D}^{(t)}$ inversely proportional to the distance of (the mean of) $\pi_t$ from $x^*$ when estimating $Q_t^{\pm}$ from $\mathcal{D}^{(\leq t)}$. We only report the vanilla versions described above in our experiments and leave further variance reduction techniques for future work. In all experiments, we use $K = n$ and the Euclidean distance on $\mathcal{X}$.

While these locally guided strategies appear rather limited in which information from previous rounds is used in updating the policy, Singh et al. (2019, Sec. 7) make a convincing case for why the simple types of active learning in EE and AEE greatly improve over the naive random strategy and may be competitive with fully adaptive experiments. For the fixed policy in $\pi$ and the meta-distribution $\Gamma$ one should arguably err on the side of 'uninformative' priors, i.e., covering all of $\mathcal{Z}$ mostly uniformly.

**Targeted Adaptive Strategy.** We now develop an adaptive strategy denoted by **adaptive** that actually uses the intermediate estimates of $Q_t^{\pm}$ to update $\pi_t$. A natural choice for policy updates is via gradient descent on our objective $\Delta$

$$\pi_{t+1} = \pi_t - \alpha_t \nabla_\pi \Delta(\pi)|_{\pi=\pi_t}\,, \tag{12}$$

for step sizes $\alpha_t > 0$ at round $t$. In practice, we assume parametric policies $\pi_t := \pi_{\phi_t}$ with parameters $\phi_t \in \Phi \subset \mathbb{R}^d$. With the log-derivative trick (Williams, 1992) we then compute

$$\nabla_\phi \Delta(\pi_\phi) = \mathbb{E}_{X,Y,Z \sim P_{\pi_\phi}(X,Y,Z)} \left[ \Delta(\pi_\phi) \nabla_\phi \log \pi_\phi \right]\,. \tag{13}$$

and can update $\pi_{\phi_t}$ akin to the REINFORCE algorithm with horizon one. Since the gradient in eq. (12) is evaluated at $\pi_{\phi_t}$ from which we have samples available, we can directly use the empirical counterpart of the expectation in eq. (13). One caveat is that our objective $\Delta$ cannot be computed on the 'instance level' for individual samples $(x_i, y_i, z_i)$, but is already an estimate based on multiple samples. In practice, we therefore split the $n$ observations per round into random batches, and average over the $\Delta(\pi_\phi)$ estimate for each batch times the mean score function across the batch in eq. (13).

An extension to use data $\mathcal{D}^{(\tau)}$ from a previous round $\tau < t$ is to reweigh the term within the expectation in eq. (13) by $\pi_{\phi_t}(Z)/\pi_{\phi_\tau}(Z)$ (and then average across rounds). While this allows us to use more data for the gradient estimates, this approach does not yield an unbiased estimator and typically also increases variance due to arbitrarily small reweighing terms. In practice, we may want to only consider data from a small number of previous rounds, where we expect the policy not to have changed too much. Again, we believe our technique could benefit from further variance reduction techniques (Chandak et al., 2024), but leave these for future work.

While our formulation in eqs. (12) and (13) can be efficiently implemented for any parametric choice of $\pi_\phi$, in our experiments we choose a multivariate Gaussian mixture model (GMM) $\pi_\phi(z) = \sum_{m=1}^M \gamma_m \mathcal{N}(z; \mu_m, \Sigma_m)$ with weights $\gamma \in [0,1]^M$ with $\sum_{m=1}^M M = 1$, means $\mu_m \in \mathbb{R}^{d_z}$, and covariances $\Sigma \in \mathbb{R}^{d_z \times d_z}$ for $i \in [M]$. Hence, the parameters are $\phi = (\gamma, \mu_1, \ldots, \mu_M, \Sigma_1, \ldots, \Sigma_M)$. With the score function $\nabla_\phi \log \pi_\phi$ known analytically for GMMs, eq. (13) can be estimated efficiently (see Appendix B for details).

## 4 Experiments

**Setup.** We consider a low-dimensional setting (for visualization purpose) with $d_x = d_z = 2$ and

$$h_j(Z, U) = \alpha \cdot (\sin(Z_j)(1+U))\,, \quad f(X) = \beta \sum_{j=1}^{d_x} \exp(X_j)\sin(X_j) + U\,, \quad U \sim \mathcal{N}(0,1)\,, \tag{14}$$

where we use $\alpha = \beta = 20$. The local point of interest is $x^* = 0 \in \mathbb{R}^{d_x}$ and we easily check that $\partial_i f(x^*) = \beta$. We use $n = 250$ samples in each round over a total of $T = 16$ experiments.

**Method parameters.** We use radial basis function (RBF) kernels $k(x_1, x_2) = \exp(\rho\|x_1 - x_2\|^2)$ with a fixed $\rho = 1$. Note, that the three hyperparameters are relative weights. Thus we set $\lambda_s := \frac{\lambda_g \lambda_f}{\lambda_c} = 0.01$ and $\lambda_c = 0.04$, we refer to the Appendix C for a further comparison of different

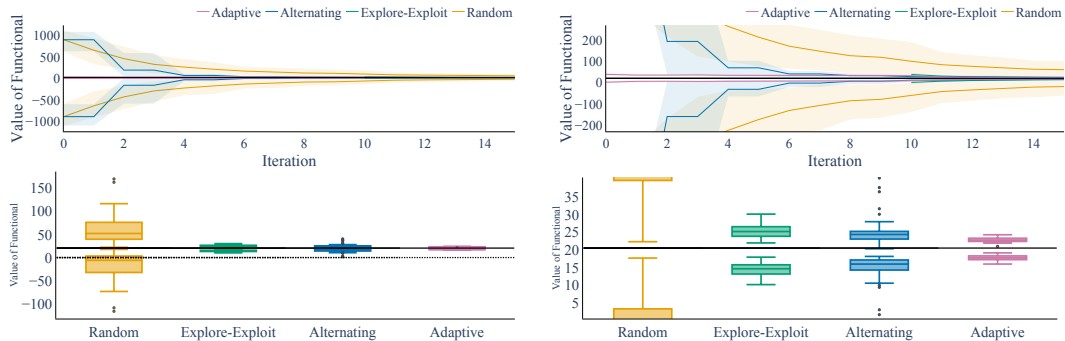

Figure 1: We compare the different strategies in our synthetic setting. Left and **right** only differ in the range of the y-axis. The black constant line represents the true value of $Q[f_0]$. **Top:** Estimated upper and lower bounds $Q_t^\pm$ over $t \in [T]$ for $n_{\text{seeds}} = 50$ and two different 'zoom levels' on the y-axis. Lines are means and shaded regions are $(10, 90)$-percentiles. **Bottom:** The final estimated bounds $Q_T^\pm$ at $T = 16$. The dotted line is $y = 0$. Both locally guided heuristic (explore-then-exploit, alternating explore exploit) confidently bound the target query away from zero with a relatively narrow gap between them. Our targeted adaptive strategy is even better and essentially identifies the target query $Q[f_0] = 20$ after $T = 16$ rounds.

hyperparameter choices. For all strategies the variance of our policy is set at $\sigma_e = 0.001$. For explore then exploit, we chose $T_1 = 10, T_2 = 6$ with $\pi_t = \mathcal{N}(\mu_t, \sigma_e \text{Id}_{d_z})$ and independent $\mu_t \sim \mathcal{N}(\mathbf{0}_{d_z}, \text{Id}_{d_z})$ for $t \in [T_1]$. The Gaussian mixture of the adaptive strategy is initialized with $M = 3, \gamma = (1/3, 1/3, 1/3), \Sigma_m = \text{Id}_{d_z}$ and independent $\mu_m \sim \mathcal{N}(\mathbf{0}_{d_z}, \text{Id}_{d_z})$. We use a constant learning rate $\alpha_t = 0.01$ for all $t$ and restrict ourselves to learning only weights $\gamma_m$, means $\mu_m$ and the diagonal entries of the covariances $\Sigma_m$.

**Results.** We perform $n_{\text{seeds}} = 50$ runs for different random seeds and report means with the 10- and 90-percentiles in Figure 1. The simple non-adaptive random baseline starts out with poor performance as expected, and improves mildly over multiple rounds. We note that the ultimate performance of the random baseline depends primarily on how much mass the random exploration (defined by the meta-distribution $\Gamma$) puts near $x^*$, i.e., whether we collect sufficiently many samples near $x^*$ over the $T$ rounds. Explore then exploit performs quite well as soon as we start exploitation. Again, whether sufficient informative examples have been collected during the exploration stage depends on the choice of $\Gamma$. However, explore then exploit does outperform random (with the same exploration distribution) indicating that the heuristic of focusing on the informative samples (the ones near $x^*$) does provide substantial improvements (as also found by (Singh, 2023)). The alternating strategy AEE provides bounds across all rounds and clearly shows step-wise improvement from the first round on—ultimately performing similar to EE. Finally, the adaptive strategy quickly narrows the bounds and achieves a fairly narrow gap $\Delta$ already after few rounds. At $T = 16$ it has essentially identified the target query $Q[f_0] = 20$ with low variance across seeds. For completeness, we provide a runtime comparison of the different methods in Appendix D. The code to reproduce results is available at https://github.com/EAiler/targeted-iv-experiments.

## 5 Discussion and Conclusion

**Summary.** We formalized designing optimal experiments to learn about a scientific query as sequential instrument design with the goal of minimizing the gap between estimated upper and lower bounds of a target functional $Q$ of the ground truth mechanism $f_0$. We only assume indirect experimental access, allow for unobserved confounding, and consider nonlinear $f_0$ with multi-variate inputs such that both $f_0$ and $Q[f_0]$ may be unidentified. For a broad set of queries, we derive closed form estimators for the bounds within each round of experimentation when $f$ lies within an RKHS. Based on these estimates, we then develop adaptive strategies to sequentially narrow the bounds on the scientific query of interest and demonstrate the efficacy of our method in synthetic experiments. Given the increased amount of data collected in fully or partially automated labs, for example for drug

discovery, we believe that efficient, adaptive experiment design strategies will be a vital component of data-driven scientific discovery.

**Limitations and future work.** We have not validated our method in a real-world setting, as it would require access (and full control) over an actual experimental setup as well as the time and resources required to conduct these experiments. A key technical limitation of our work is that neither our general policy learning formulation in eqs. (4) and (5) nor the concrete closed form estimates in Theorem 2 obtain provably valid bounds for all queries $Q$ from finite samples (see discussion right before Section 3.4). While our approach is reliable in synthetic experiments, we believe thorough theoretical analysis of necessary and sufficient conditions for (asymptotically) valid bounds (including ideally asymptotic normality with known asymptotic covariance or even finite sample guarantees) is a useful direction for future work beyond the scope of our current manuscript. In practice, the applicability of our approach may also limited by the assumptions that the underlying system is well described via a fixed, static function $f_0 : \mathcal{X} \to \mathcal{Y}$ as opposed to, say, a temporally evolving and interacting systems governed by a differential equation. Similarly, while IV assumptions (1) and (2) can arguably be justified in our setting, the third assumption $Z \perp\!\!\!\perp Y \mid X, U$ limits the types of experimentation we can consider (see discussion at the end of Section 1).

Methodologically, we believe that our approach could benefit from advanced variance reduction techniques and be sped up by more efficient estimators (Chandak et al., 2024). Along these lines, it is worthwhile future work to analyze the optimal trade-off between exploration and retaining a large number of samples for exploitation and thus lower variance estimates. Finally, our current empirical validation is limited to linear local functionals, rather simple parametric choices for policies (such as (mixtures of) Gaussians), and we have not optimized the kernel choices and hyperparameters. Hence, a validation where all these choices have been tailored to a real-world application scenario is an important direction for future work.

## Acknowledgments and Disclosure of Funding

EA is supported by the Helmholtz Association under the joint research school "Munich School for Data Science – MUDS". This work has been supported by the Helmholtz Association's Initiative and Networking Fund through the Helmholtz international lab "CausalCellDynamics" (grant # Interlabs-0029).

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

# A Proofs

In this section we provide proofs of the statements in the main paper. Before proving Theorem 2 we restate a result from the literature that provides closed form consistent estimates of the minimax problem with the $L_2$ penalty term (instead of penalizing the functional).

**Theorem 3** (Proposition 10 of (Dikkala et al., 2020)). *Assume without loss of generality that functions in $\mathcal{H}_Z, \mathcal{H}_X$ have bounded variation and their images are contained in $[-1, 1]$. Additionally, assume that for each $h \in \mathcal{H}_Z$ also $-h \in \mathcal{H}_Z$. Denote by $K_{XX}$ and $K_{ZZ}$ the empirical kernel matrices from $\mathcal{D}$. Then we can consistently estimate*

$$f = \arg\inf_{f \in \mathcal{F}} \sup_{g \in \mathcal{G}} \mathbb{E}[(Y - f(X))g(Z)] - \lambda_g \|g\|_{\mathcal{G}} + \lambda_f \|f\|_{\mathcal{F}} \tag{15}$$

*via*

$$\hat{f}(\cdot) = \sum_{i=1}^{n} \hat{\theta}_i k(X_i, \cdot) \qquad \hat{\theta} = (K_{XX} M K_{XX} + 4\lambda_g \lambda_f K_{XX})^+ K_{XX} My \tag{16}$$

*with $M = K_{ZZ}$. If the function classes $\mathcal{F}, \mathcal{G}$ are already norm constrained, the estimator needs no penalization.*

*Proof.* Note that we have the optimum of the inner supremum

$$\sup_{g \in \mathcal{G}} \mathbb{E}[(Y - f(X))g(Z)] - \lambda_g \|g\|_{\mathcal{G}} \tag{17}$$

at

$$\frac{1}{4\lambda_g}(Y - f(X))^\top M(Y - f(X)) \tag{18}$$

Therefore, we are left to solve the outer minimization of

$$\hat{f} = \min_{f \in \mathcal{F}}(Y - f(X))^\top M(Y - f(X)) + 4\lambda_g \lambda_f \|f\|_{\mathcal{F}}^2 \tag{19}$$

$$\hat{f} = \min_{f \in \mathcal{F}}(Y - f(X))^\top M(Y - f(X)) + 4\lambda_g \lambda_f \|f\|_{\mathcal{F}}^2$$

$$\min_{\theta \in \mathbb{R}^n}(y - K_{XX}\theta)^\top M(y - K_{XX}\theta) + 4\lambda_g \lambda_f \|K_x\theta\|_2^2 \qquad \left| \text{Replacing the function with the (empirical) kernels} \right.$$

$$\min_{f \in \mathcal{F}} \theta^\top K_{XX} M K_{XX}\theta - 2y^\top M K_{XX}\theta + 4\lambda_g \lambda_f \theta^\top K_x\theta \qquad \left| \text{Writing out product} \right.$$

$$\min_{f \in \mathcal{F}} \theta^\top (K_{XX} M K_{XX} + 4\lambda_g \lambda_f K_{XX})\theta - 2y^\top M K_{XX}\theta \qquad \left| \text{Aggregation of terms wrt } \theta \right.$$

$$\hat{f} = (K_{XX} M K_{XX} + 4\lambda_g \lambda_f K_{XX})^+ K_{XX} My \qquad \left| \text{Solution for convex minimization problems} \right.$$

$\square$

**Theorem 2.** *Assume functions in $\mathcal{H}_Z, \mathcal{H}_X$ to have bounded variation, (w.l.o.g.) images contained in $[-1, 1]$, and for $h \in \mathcal{H}_Z$ also $-h \in \mathcal{H}_Z$. Denote by $K_{XX}, K_{ZZ} \in \mathbb{R}^{n \times n}$ the empirical kernel matrices of $\mathcal{D}$, let $k_X$ be continuously differentiable, and let $\lambda_g, \lambda_f, \lambda_c \geq 0$. Further, fix $Q[f]$ to be a bounded linear functional, and write $f_\theta(\cdot) = \sum_{i=1}^n \theta_i k(x_i, \cdot)$. Then the solutions to the regularized minimax problems*

$$f^\pm = \arg\min_{f \in \mathcal{H}_X} \mp Q[f] + \lambda_c \sup_{g \in \mathcal{H}_Z} \langle r_0 - Tf, g \rangle_{\mathcal{H}_Z} + \lambda_f \|f\|_{\mathcal{H}_X} - \lambda_g \|g\|_{\mathcal{H}_Z} \tag{8}$$

*are consistently estimated by $f_{\hat{\theta}^\pm}$ with*

$$\hat{\theta}^\pm = (K_{XX} K_{ZZ} K_{XX} + 4\frac{\lambda_g \lambda_f}{\lambda_c} K_{XX})^+ \left( K_{XX} K_{ZZ} y \mp \frac{1}{\lambda_c} Q[K_{X\cdot}](x^*) \right), \tag{9}$$

*where $(Q[K_{X\cdot}])(x^*) \in \mathbb{R}^n$ is the vector $((Q[k_X(x_1, \cdot)])(x^*), \ldots, (Q[k_X(x_n, \cdot)])(x^*)$. Note that these general bounded linear functionals can be computed analytically for many common kernels such as linear, polynomial, or radial basis function (RBF) kernels.*

*Proof.* The solution of the inner supremum remains the same as in Theorem 3. This leaves us to solve the outer minimization

$$\min_{f \in \mathcal{H}_X} Q[f] + 4\lambda_g\lambda_f\|f\|_{\mathcal{H}_X} + \lambda_c(y - K_{XX}\theta)^\top M(y - K_{XX}\theta), \tag{20}$$

where $M = K_{ZZ}$. We write out $f$ as a linear combination of kernel basis functions (at the observed data points) and use the linearity of the gradient functional, which applied to the kernel basis functions is just $(\partial_i K_X.)(x^*)$. This yields for the overall functional $Q[f_\theta] = (\partial_i K_X.)(x^*)\theta$. We are thus seeking

$$\min_{\theta \in \mathbb{R}^n} (\partial_i K_X.)(x^*)\theta + \lambda_c\theta^\top K_{XX}MK_{XX}\theta - 2\lambda_c y^\top MK_{XX}\theta + 4\lambda_g\lambda_f\theta^\top K_{XX}\theta, \tag{21}$$

which we rewrite as

$$\min_{\theta \in \mathbb{R}^n} \theta^\top \left( K_{XX}MK_{XX} + 4\frac{\lambda_g\lambda_f}{\lambda_c}K_{XX} \right)\theta + \left( \frac{1}{\lambda_c}(\partial_i K_X.)(x^*) - 2y^\top MK_{XX} \right)\theta. \tag{22}$$

Building on the arguments in the proof of Theorem 3 where we replace the linear part in $\theta$ by the partial derivative functional, we obtain the minimizer

$$\left( K_{XX}MK_{XX} + 4\frac{\lambda_g\lambda_f}{\lambda_c}K_{XX} \right)^+ \left( K_{XX}My - \frac{1}{\lambda_c}(\partial_i K_X.)(x^*) \right) \tag{23}$$

as a consistent estimator. $\qquad\square$

*Proof.* We re-iterate the same arguments to prove the theorem for general bounded linear functionals. We assume $Q[f]$ to be a linear bounded functional acting on $f$.

**Theorem 4.** *Riesz representation theorem* *Let $Q[f] : \mathcal{H}_X \to \mathbb{R}$ be a continuous linear functional on a separable Hilbert space $\mathcal{H}_X$. Then there exists a $q \in \mathcal{H}_X$ such that $Q[f] = \langle f, q \rangle_{\mathcal{H}_X}, \forall f \in \mathcal{H}_X$.*

Assuming $f \in \mathcal{H}_X$, via the Riesz-representer theorem for linear functionals, there exists a unique element $q \in \mathcal{H}_X$ such that for any $f \in \mathcal{H}_X$. The bounded linear functionals on $\mathcal{H}_X$ form themselves a hilbert space called the dual space.

$$Q[f] = \langle f, q \rangle_{\mathcal{H}_X} \tag{24}$$

We assume $\mathcal{H}_X$ to be an RKHS. Therefore, we can write out $f(x)$ as a linear combination of the kernel basis functions (at the observed data points):

We replace $Q[f] = \langle f, q \rangle_{\mathcal{H}_X} = \langle \sum_{i=1}^n \theta_i k(x_i, \cdot), q \rangle = \sum_{i=1}^n \theta_i q(x_i)$ as $q \in \mathcal{H}_X$. Due to the RKHS, we can write the empirical version as $q(\cdot) = \sum_{i=1}^n Q[k(x_i, \cdot)]$.

$$\min_{f \in \mathcal{H}_X} \theta^\top Q[k(x_i, \cdot)] + 4\lambda_g\lambda_f\theta^\top K_{XX}\theta + \lambda_c(y - K_{XX}\theta)^\top M(y - K_{XX}\theta) \tag{25}$$

Thus we obtain the minimizer

$$\left( K_{XX}MK_{XX} + 4\frac{\lambda_g\lambda_f}{\lambda_c}K_{XX} \right)^+ \left( K_{XX}My - \frac{1}{\lambda_c}Q[K_X.(x^*)] \right) \tag{26}$$

$\qquad\square$

# B  Details on Adaptive Policies

For completeness, we here recall the score function of Gaussian mixture models, i.e., the gradients with respect to the mixture weights, means, and covariances of the log-likelihood function. We write $\pi$ for the GMM for brevity. First, we define the 'responsibilities' $\zeta_{im}$, the probability that a sample $z_i$ comes from component $m$ via (see, e.g., Petersen & Pedersen (2008))

$$\zeta_{im} = \frac{\gamma_m\mathcal{N}(z_i \mid \mu_m, \Sigma_m)}{\sum_{j=1}^M \gamma_j\mathcal{N}(z_i \mid \mu_j, \Sigma_j)}. \tag{27}$$

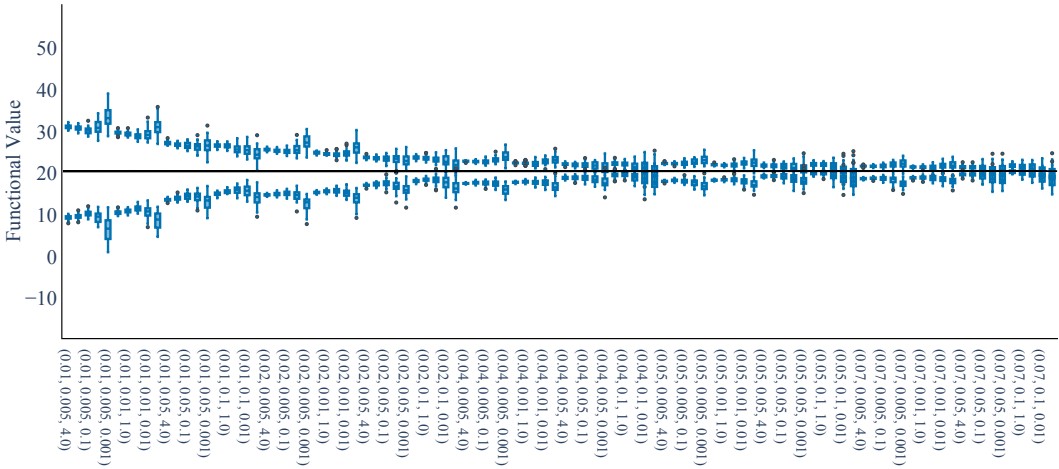

Figure 2: Adaptive Method for different hyperparameter settings. The x-axis shows them in the following order: $(\lambda_c, \lambda_s, \alpha_t)$

Then, we have for a sample $z_i$ and mixture component $m \in [M]$ that

$$\nabla_{\gamma_m} \log \pi(z_i) = \frac{\zeta_{im}}{\gamma_m} \,, \tag{28}$$

$$\nabla_{\mu_m} \log \pi(z_i) = \zeta_{im} \Sigma_m^{-1}(z_i - \mu_m) \,, \tag{29}$$

$$\nabla_{\Sigma_m} \log \pi(z_i) = -\frac{1}{2}\zeta_{im}(\Sigma_m^{-1} - \Sigma_m^{-1}(z_i - \mu_m)(z_i - \mu_m)^T \Sigma_m^{-1}) \,. \tag{30}$$

## C   Hyperparameter Tuning

In Eq. (11) we end up with three hyperparameters. Note however, that those three mentioned hyperparameters are relative weights. Thus, we can set $\lambda_s := \frac{\lambda_f \lambda_g}{\lambda_c}$ and only tune $\lambda_s$ and $\lambda_c$. Intuitively, $\lambda_s$ regularizes the smoothness of the function spaces and $\lambda_c$ weighs the functional $Q$ relative to the moment conditions. In Fig. 2, we show the dependence of our bounds on these hyperparameters: very low values of $\lambda_s$ lead to conservative bounds whereas large values of $\lambda_s$ yield narrow bounds throughout, as $\lambda_s$ effectively controls the size of the search space via regularity restrictions. For the learning rate, a lot of learning rates below a certain value actually work for the method. More gradient update steps (and therefore more experiments) are required for very small learning rates, very high learning rates result in drastic updates of the policy; something that is not very favorable in real-world experiments.

## D   Runtimes

For completeness, we show wallclock runtimes of the different methods in our empirical evaluation in Figure 3. All methods were run on a MacBook Pro with Intel CPU. The runtimes are per iteration and clearly increase for methods that accumulate data over previous rounds. Even the most expensive passive baseline is relatively fast to compute without specialized hardware and we generally imagine computational cost to be negligible compared to the actual cost of physical experimentation required in each round in most applications.

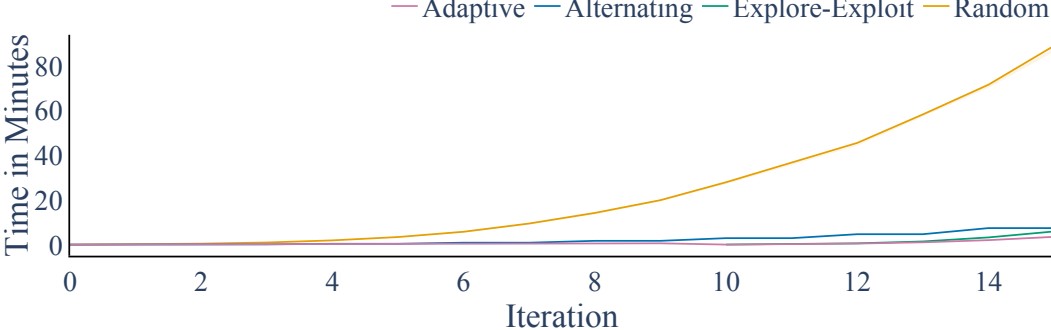

Figure 3: Wallclock runtimes of the different methods.

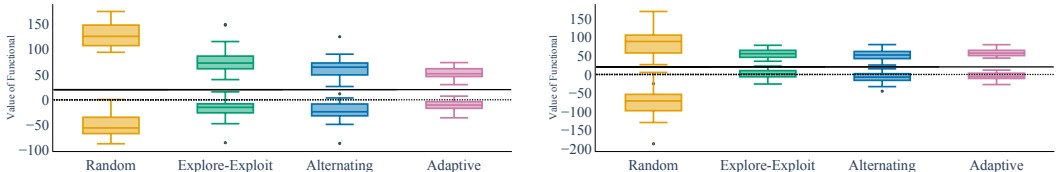

Figure 4: We compare the different strategies in our synthetic setting. The black constant line represents the true value of $Q[f_0]$. Both plots show the estimated upper and lower bounds $Q_t^\pm$ at $T = 16$ for $n_{\text{seeds}} = 50$. Lines are means and shaded regions are $(10, 90)$-percentiles. **Left:** $d_z = 5, d_x = 20$, **Right:** $d_z = d_x = 20$.

## E   Additional Experiments

We assume the same setting as the low-dimensional one, only increasing the dimensions.

$$h_j(Z, U) = \begin{cases} \alpha \cdot \sin(Z_j) \cdot (1 + U), & \text{if } j \leq d_z, \\ 1 + U, & \text{if } j > d_z. \end{cases} \tag{31}$$

$$f(X) = \beta \sum_{j=1}^{d_x} \exp(X_j) \sin(X_j) + U, \quad U \sim \mathcal{N}(0, 1), \tag{32}$$

with $d_z = 5, d_x = 20$ and $d_z = 20, d_x = 20$. The setting guarantees, that though we have a mismatch in dimensions, i.e. $d_z < d_x$, we would—in principle—still be able to identify the full causal effect, as the completeness property is fulfilled. We refer to Fig. 4 for the results of the different methods. Moreover, we note that the increase in time is still manageable, c.f. Fig 5. The main driver for the computation time are the samples within each experiment $(T_1, T_2)$ due to the kernel approach and, therefore, the inversion of gram matrices based on the sample size.

For $d_z = 5, d_x = 20$, we used $\lambda_s := \frac{\lambda_g \lambda_f}{\lambda_c} = 0.04$ and $\lambda_c = 0.1$. For $d_z = d_x = 20$, we used $\lambda_s := \frac{\lambda_g \lambda_f}{\lambda_c} = 0.05$ and $\lambda_c = 0.1$. Intuitively, the optimization requires stronger hyperparameters in higher dimensional settings as with dimensions, also the function space increases.

## F   Underspecification

Additionally, we include some examples along the lines of underspecification. We refer to Ailer et al. (2023) for a thorough discussion on underspecification, but give some motivation for the examples in the following:

Since $f$ is the solution to a linear inverse problem, the problem is often ill-posed and thus $f$ is not identified. However, underspecification means something more explicit. It can be seen as a violation of the completeness property.

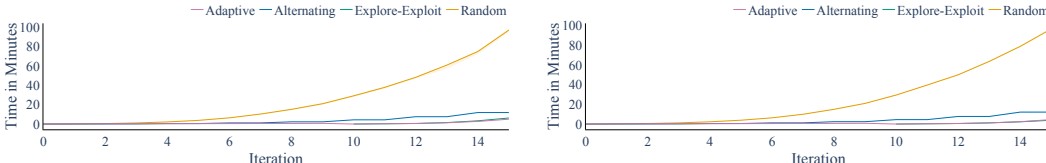

Figure 5: We compare the wallclock time in a higher dimensional setting. The time increase is still mild. The main driver is the number of samples in each experiment, instead of the dimensionality itself. **Left:** $d_z = 5, d_x = 20$, **Right:** $d_z = d_x = 20$.

**Lemma 5** (Severini & Tripathi (2006))**.** *The conditional distribution of $p(X \mid Z)$ is complete if and only if for each function $f(x)$ such that $\mathbb{E}[f(x)] = 0$ and $Var(f(x)) > 0$, there exists a function $g(z)$ such that $f(x)$ and $g(z)$ are correlated.*

In most papers, the completeness property is trivially fulfilled by assuming an exponential family for $p(x \mid z)$ with non-zero variance for all components. This means that both $T$ and $T^*$ are injective which again guarantees the identifiability of $f$.

Now, we denote $\mathcal{P}_{\mathcal{F}}$ as the orthogonal projection onto the function space $\mathcal{F}$ and $\mathcal{N}(T)$ as the null space of the linear operator $T : \mathcal{H}_X \to \mathcal{H}_Z$ with $Tf = T[f(X) \mid Z]$. Following this notation, Severini & Tripathi (2012) argue that $\mathcal{P}_{\mathcal{N}(T)^{\perp}} f$ is the identifiable part of $f$. This leads to the following lemma:

**Lemma 6** (Severini & Tripathi (2012))**.** $\mathbb{E}[Q[f]]$ *is identified if and only if $Q \in \mathcal{N}(T)^{\perp}$.*

**Examples.** Let us introduce an example for an unidentified $Q[f]$: Consider $Z, Y \in \mathbb{R}$, $X \in \mathbb{R}^2$ and $h(z) = (z, 0)$, $f(x_1, x_2) = 0.5x_1 + 2x_2$ and $Q[f] = \partial_2 f(x^*)$. In this fully linear setting, due to the structure of $h$ the instrument can only ever perturb the first component of $X$ regardless of what policy we choose. Hence, it is impossible to identify e.g. $Q[f]$ w.r.t. the second argument with $\partial_2 f(x^*) = 2 \forall x^*$. Even a policy with full support will not (even partially) identify $Q[f_0]$. In contrast, let us discuss a fully identifiable scenario, which is identifiable for all $Q[f]$, but still depends on the policy: Consider a similar setting with $f(x_1, x_2) = x_1^2 + x_2$ and $Q[f] = \partial_1 f(x^*)$. Assume we have $x^* = (6, 1)$, then $Q[f_0] = 2 \cdot 6 = 12$. Any policy $\pi$ that has positive density in a neighborhood of $z = 6$ will be able to fully identify $Q[f_0]$. However, any policy that puts no mass (or in practice, little mass) near $z = 6$, will not identify $Q[f_0]$. This would be an uninformative policy.

# G  Algorithmic Boxes

We present pseudocode for the 'explore then exploit' (EE) strategy in Algorithm 1, for 'alternating explore exploit' (AEE) in Algorithm 2, and for the adaptive strategy in Algorithm 3.

---

**Algorithm 1** Explore then exploit (EE)

---

1: **Input:** Budget $T$, $T_1$, $T_2$, $n$, distance measure $d$ on $\mathcal{X}$, target $x^*$
2: **Initialize:** $\mathcal{D}^{(\leq T_1)} \leftarrow \emptyset$
3: Split budget: $T = T_1 + T_2$
4: **for** $t = 1$ to $T_1$ **do**
5:     Sample $\pi_t \sim \Gamma$
6:     Observe sample $(X_t, Z_t)$
7:     $\mathcal{D}^{(\leq T_1)} \leftarrow \mathcal{D}^{(\leq T_1)} \cup \{(X_t, Z_t)\}$
8: **end for**
9: Find $K$ nearest samples to $x^*$ in $\mathcal{D}^{(\leq T_1)}$ based on distance $d$
10: Fit $\pi$ as a parametric distribution on the $Z$ values of these $K$ samples
11: **for** $t = T_1 + 1$ to $T$ **do**
12:     Set $\pi_t := \pi$
13:     Observe sample $(X_t, Z_t)$
14:     $\mathcal{D}^{(t)} \leftarrow \{(X_t, Z_t)\}$
15: **end for**
16: Estimate $Q_t^{\pm}$ from $\bigcup_{t=T_1+1}^{T} \mathcal{D}^{(t)}$

---

---

**Algorithm 2** Alternating explore exploit (AEE)

---

1: **Input:** Budget $T$, $K$, $n$, distance measure $d$ on $\mathcal{X}$, target $x^*$
2: **Initialize:** Sorted list of observed samples $\mathcal{L} \leftarrow \emptyset$
3: **for** $t \in [T]$ **do**
4:     **if** $t$ is odd and $t \neq T$ **then**
5:         Sample $\pi_t \sim \Gamma$
6:         Observe sample $(X_t, Z_t)$
7:         $\mathcal{L} \leftarrow \mathcal{L} \cup \{(X_t, Z_t)\}$
8:         Sort $\mathcal{L}$ by distance $d(X, x^*)$
9:     **else if** $t$ is even or $t = T$ **then**
10:         Find the $\min\{K, n \cdot t\}$ nearest samples to $x^*$ in $\mathcal{L}$
11:         Fit $\pi_t$ as a parametric distribution on the $Z$ values of these samples
12:         Observe sample $(X_t, Z_t)$
13:         $\mathcal{L} \leftarrow \mathcal{L} \cup \{(X_t, Z_t)\}$
14:         Sort $\mathcal{L}$ by distance $d(X, x^*)$
15:     **end if**
16: **end for**
17: Estimate $Q_t^{\pm}$ on $\bigcup_{\text{even } t} \mathcal{D}^{(t)}$

---

---

**Algorithm 3** Adaptive

---

1: **Input:** Budget $T$, $T_1$, $T_2$, $n$
2: **Initialize:** $\Phi_t$
3: Split budget: $T = T_1 + T_2$
4: **for** $t = 1$ to $T_1$ **do**
5:     Sample $Z_t \sim \pi_{\Phi_t}$
6:     Observe sample $(X_t, Z_t)$
7:     Compute gradient update $\nabla(\Phi_t)\Delta(\pi_{\Phi_t})$ by splitting observed samples
8: **end for**
9: **for** $t = T_1 + 1$ to $T$ **do**
10:     Set $\pi_t := \pi_{\Phi_{T_1}}$
11:     Observe sample $(X_t, Z_t)$
12:     $\mathcal{D}^{(t)} \leftarrow \{(X_t, Z_t)\}$
13: **end for**
14: Estimate $Q_t^{\pm}$ from $\bigcup_{t=T_1+1}^{T} \mathcal{D}^{(t)}$

---


Table 1: Overview of resources used in our work.

| Name | Reference | License |
|---|---|---|
| Python | Van Rossum & Drake (2009) | PSF License |
| PyTorch | Paszke et al. (2019) | BSD-style license |
| Numpy | Harris et al. (2020) | BSD-style license |
| Pandas | pandas development team (2020) | BSD-style license |
| Jupyter | Kluyver et al. (2016) | BSD-style license |
| Matplotlib | Hunter (2007) | modified PSF (BSD compatible) |
| Scikit-learn | Pedregosa et al. (2011) | BSD 3-Clause |
| SciPy | Virtanen et al. (2020) | BSD 3-Clause |
| JAX | Bradbury et al. (2018) | |
| SLURM | Andy et al. (2003) | Apache 2.0 |

Guidelines:

- The answer NA means that the paper does not use existing assets.
- The authors should cite the original paper that produced the code package or dataset.
- The authors should state which version of the asset is used and, if possible, include a URL.
- The name of the license (e.g., CC-BY 4.0) should be included for each asset.
- For scraped data from a particular source (e.g., website), the copyright and terms of service of that source should be provided.
- If assets are released, the license, copyright information, and terms of use in the package should be provided. For popular datasets, `paperswithcode.com/datasets` has curated licenses for some datasets. Their licensing guide can help determine the license of a dataset.
- For existing datasets that are re-packaged, both the original license and the license of the derived asset (if it has changed) should be provided.

- If this information is not available online, the authors are encouraged to reach out to the asset's creators.

13. **New Assets**

   Question: Are new assets introduced in the paper well documented and is the documentation provided alongside the assets?

   Answer: [Yes]

   Justification: While our main contribution is not the code that we provide alongside the paper, but only provided to ensure reproducibility, the code is still well documented and ready to use. The consent is given, as the authors of the papers are owners of the code as well.

   Guidelines:
   - The answer NA means that the paper does not release new assets.
   - Researchers should communicate the details of the dataset/code/model as part of their submissions via structured templates. This includes details about training, license, limitations, etc.
   - The paper should discuss whether and how consent was obtained from people whose asset is used.
   - At submission time, remember to anonymize your assets (if applicable). You can either create an anonymized URL or include an anonymized zip file.

14. **Crowdsourcing and Research with Human Subjects**

   Question: For crowdsourcing experiments and research with human subjects, does the paper include the full text of instructions given to participants and screenshots, if applicable, as well as details about compensation (if any)?

   Answer: [NA]

   Justification: The paper does not involve any crowdsourcing or research with human objects.

   Guidelines:
   - The answer NA means that the paper does not involve crowdsourcing nor research with human subjects.
   - Including this information in the supplemental material is fine, but if the main contribution of the paper involves human subjects, then as much detail as possible should be included in the main paper.
   - According to the NeurIPS Code of Ethics, workers involved in data collection, curation, or other labor should be paid at least the minimum wage in the country of the data collector.

15. **Institutional Review Board (IRB) Approvals or Equivalent for Research with Human Subjects**

   Question: Does the paper describe potential risks incurred by study participants, whether such risks were disclosed to the subjects, and whether Institutional Review Board (IRB) approvals (or an equivalent approval/review based on the requirements of your country or institution) were obtained?

   Answer: [NA]

   Justification: The paper does not involve any crowdsourcing or research with human objects.

   Guidelines:
   - The answer NA means that the paper does not involve crowdsourcing nor research with human subjects.
   - Depending on the country in which research is conducted, IRB approval (or equivalent) may be required for any human subjects research. If you obtained IRB approval, you should clearly state this in the paper.
   - We recognize that the procedures for this may vary significantly between institutions and locations, and we expect authors to adhere to the NeurIPS Code of Ethics and the guidelines for their institution.
   - For initial submissions, do not include any information that would break anonymity (if applicable), such as the institution conducting the review.

