# OpenReview forum: "Targeted Sequential Indirect Experiment Design"
_NeurIPS.cc/2024/Conference — NeurIPS 2024 poster_

### Official Review · Reviewer_sEkT · 2024-07-09

**Soundness:** 2
**Presentation:** 3
**Contribution:** 3
**Rating:** 5
**Confidence:** 3

**Summary:**

This paper designs comprehensive experiments that maximize the information gained about the query of interest within a fixed budget of experimentation, including nonlinear, multi-variate, confounded settings.

**Strengths:**

This paper is well-motivated and relevant to the causal inference. Overall, this work is well-written and organized.

**Weaknesses:**

1. Are the theoretical results practically useful when addressing real-world problems?
2. How does the proposed method handle non-convex optimization, and what are the assumptions on the loss function?
3. Can the authors provide more intuitive theoretical explanations of the difference between convex and non-convex optimization in their settings?
4. How can the effectiveness of the proposed method be guaranteed when the dimension p is significantly larger than the number of training samples?

**Questions:**

1. Are there any crossing problems when minimizing the optimization problem in Equation (5)? How do we ensure that $Q^{+}(\pi)$ is always larger than $Q^{-}(\pi)$?
2. Theoretically, are there any requirements for the dimensions of $Z$ and $U$? In your experiments, you only consider a low-dimensional setting. Can you add any experiments for high dimensions for $Z$ and $U$?
3. For Theorem 2, the proposed method involves several tuning parameters, such as $λ_g,λ_f,λ_c$. This step can be quite worrying for the practical implementation of the proposed method. 1) it can be time-consuming and 2) setting the candidate values is likely quite subjective. The authors should conduct more thorough studies and provide more guidelines on how the choice of tuning parameters affects the method's performance. For example, the authors have selected $λ_g=0.1,λ_f=0.1,λ_c=0.1$, and the learning rate $α_t=0.01$. This selection process appears too subjective.
4. The experiments are not sufficient for several reasons: (1) The number of replications used in the experiments is quite small (only 25), which raises concerns about the time efficiency of the proposed method. (2) The paper does not include a real-world data analysis to validate the effectiveness of the proposed algorithm from a practical perspective.

**Limitations:**

1. Theoretical requirements for the dimensions of $Z$ and $U$ are not discussed. Experiments only consider a low-dimensional setting, see questions
2. The proposed method involves several tuning parameters, which can be problematic for practical implementation due to time consumption and the subjective setting of candidate values; see questions.
3. Insufficient Experiments: see questions.

---

> ### Author Rebuttal · Authors · 2024-08-07
>
> We thank the reviewer for their useful comments and address the mentioned concerns/questions one by one.
>
> ## Weaknesses
>
> 1 **Practical usefulness**: Yes, we do believe our contribution is practically useful. In particular, academia (e.g., the “A-Lab” at the Berkeley lab) and industry (big pharma and biotech startups) alike work feverishly on AI driven “automated labs” to accelerate drug, material, or molecule design and scientific discoveries. Efficient and targeted adaptive algorithms to inform subsequent experiments remain a major open challenge. Notably, our targeted formulation (aiming at a specific property of the mechanism) and allowing for non point-estimable properties (via bounding) are major steps towards this goal.
>
> 2+3 **Convexity**: We are not entirely sure which (non-)convexity is being referred to, and appreciate a clarification if we are answering the wrong question.
> There are multiple optimizations. First, there is the minimax problem (eq. (4)) required to obtain a single upper/lower bound on $Q[f_0]$ under a fixed policy. Thm 1 and Thm 2 show that both the inner supremum (eq. (7)) as well as the outer minimization (eq. (9)) can be solved analytically within RKHSs. Hence, we do not see any issues around (non-)convexity for those. The remaining optimization is to update the policy $\pi$ from one round to the next to minimize $\Delta (\pi)$. This is an optimization over the space of distributions on $Z$ with the loss function being given by eq. (9) and estimated from finite data in practice. Depending on how we parameterize $\pi$ in practice and what we choose for $Q$, this loss function need not be convex in the optimization parameters and we optimize it with a local gradient-based method with no guarantees of finding a global optimum. Importantly, the upper/lower bounds estimation at each step is consistent so our method always provides (narrowing) valid bounds. They may just not be as narrow as they could have been. In practice, we do achieve informative bounds throughout.
>
> 4 **p > n:** We assume that $p$ refers to $d_x$, the treatment dimension? We are relying on kernel ridge regression for the estimation tasks from $X$ to $Y$, i.e., mapping $\mathbb{R}^{d_x} \to \mathbb{R}$. Since intuitively this means that the $d_x$ features are embedded into an infinite-dimensional feature space, regularization is required in any case to deal with the “sparsity” (more features than samples), which is why we use kernel ridge. Increasing $d_x$ beyond the sample size $n$ is thus no issue in our framework and taken into account by the regularization.
>
> ## Questions
>
> 1 **Crossing:** The optimization problem in eq. 4 is the Lagrangian multiplier formulation of maximizing/minimizing $Q[f]$ over$\mathcal{F}$ subject to the moment conditions ($\mathbb{E}[(Y - f(X))g(Z)]$). Therefore, $Q^+$ and $Q^-$ are the maximum and minimum over the same set $\mathcal{F}_c$. Therefore, $Q^+ \ge Q^-$ (no crossing) is guaranteed. In practice, our estimators (inevitably) have non-zero finite sample variance, which can lead to crossing only when (a) the gap is already very small (we have essentially identified $Q[f_0]$ already), or (b) the sample size is very small (diagnosable from large variance in estimates across rounds).
>
> 2 **Dimensionality of $U$ and $Z$:** In general, there are no theoretical requirements on the dimensionality of $U$ and $Z$. The IV setting allows for arbitrary confounding $U$ of any dimension between $X$ and $Y$ and we never explicitly need $U$ in our framework (as it is unobserved). Typically a large number of instruments $Z$ is desirable as having more instruments helps identification. We primarily focus on the more difficult (yet more realistic) setting where $d_z < d_x$ and thus underspecification is likely. However, our framework equally applies to $d_z \ge d_x$, where identification is generally going to be easier. (The full theoretical conditions for identification are Sec 3.1.)
> Following the suggestion, Figs. 1 and 2 in the rebuttal pdf show results for higher-dimensional settings with $d_z = 5, d_x = 20$ and $d_z=20, d_x=20$, demonstrating that our method indeed scales to such settings while remaining informative. In the $d_z=5, d_x=20$ setting, our adaptive approach is the only one that essentially rules out negative values and the causal MSE is smallest both on average as well as in variance for our adaptive procedure.
> Fig. 3 in the rebuttal pdf reports runtimes, demonstrating the computational feasibiilty. The number of samples per experiment is the key driver of computational cost, which can be easily reduced by deploying established techniques to handle the inversion of large kernel matrices [1, 2]. We note that computational cost and time are likely negligible compared to the cost and time of physical experiments in real-world applications.
>
> 3 **Hyperparameter Tuning:** Since the three mentioned hyperparameters (the different $\lambda_s$s) are relative weights, we can set $\lambda_s := \frac{\lambda_f * \lambda_g}{ \lambda_c}$ and only tune $\lambda_s$ and $\lambda_c$. Intuitively, $\lambda_s$ regularizes the smoothness of the function spaces and $\lambda_c$ weighs the functional $Q$ relative to the moment conditions. In Fig.  4 in the pdf, we show the dependence of our bounds on these hyperparameters: very low values of $\lambda_s$  lead to conservative bounds whereas large values of $\lambda_s$ yield narrow bounds throughout, as $\lambda_s$ effectively controls the size of the search space via regularity restrictions. We note that there are substantial ranges of reasonable hyperparameter choices where our bounds are fairly insensitive to the exact choice.
> In practice, all learning rates below a certain value work. More gradient update steps are required for convergence for smaller learning rates and – as in most (stochastic) gradient based optimizations – one chooses a learning rate just below a value that yields reliable convergence.

---

> ### Author Response · Authors · 2024-08-07
>
> 4 **Insufficient experiments:** Our runtime evaluation in appendix C, Fig. 4 (see also Fig. 3 in the rebuttal pdf), shows that the computational cost of our approach is not a severe limitation, i.e., runtime is not a real concern. While 25 replications already provide a good assessment of the finite sample variance, we happily increase this number to 100 for the revised version. From the first finished settings, the results are visually unchanged.
> We agree that we would have loved to include a real-world application of the method, but point to our discussion of the fundamental difficulty of assessing our method in a real-world setting in our “limitations” setting (l.392 and following).

---

> > ### Comment · Reviewer_sEkT · 2024-08-14
> >
> > Thanks for the author's response, which addressed most of my concerns. I will maintain my score.

---

### Official Review · Reviewer_aNmJ · 2024-07-12

**Soundness:** 3
**Presentation:** 3
**Contribution:** 3
**Rating:** 6
**Confidence:** 3

**Summary:**

This paper proposes a framework for designing sequential indirect experiments for estimating targeted scientific queries in complex, nonlinear environments with potential unobserved confounding when direct intervention is impractical or impossible. The authors formulate the problem as the sequential instrument design, using instrumental variables and minimax optimization to estimate the upper and lower bounds of targeted causal effects. The proposed method then tightens these bounds iteratively through adaptive strategies. Experiments on simulated data demonstrate the proposed method's efficacy compared to non-adaptive experimental design baselines.

**Strengths:**

1. This paper focuses on targeted sequential indirect experimental design, which is an important problem in scientific discovery where direct intervention is often impractical or impossible.

2. The proposed method is more flexible as it considers nonlinear, multi-variate, and confounded settings, and formulating the problem as a sequential underspecified instrumental variable estimation is intuitive and sound.

3. The authors develop closed-form estimators for the bounds given targeted queries when the mechanism is in an RKHS.

4. The proposed method outperforms non-adaptive baselines in synthetic experiments.

**Weaknesses:**

1. Although the proposed method focuses on indirect experimental design, it still requires the causal structure between variables known, which is also one of the main challenges in scientific discovery. I am curious about the method's sensitivity to imperfections in the causal structure, assumptions regarding instrumental variables, and the presence of confounders. It would be great if the authors could discuss further about them.

2. The authors only conduct synthetic experiments in a simple setting when the number of variables is small. I wonder if the authors could conduct more experiments in a more complex setting with a larger number of nodes, and also discuss the scalability of the proposed method (since kernel-based techniques are used for optimization as mentioned in the paper).

**Questions:**

1. Please see the questions in the Weaknesses part.

2. It seems that the proposed method is closely related to causal Bayesian Optimization [1]. I wonder if the authors could discuss the connection in detail.

3. Experiments demonstrate that the proposed method performs well for local causal effect queries. However, it is usually unknown whether the target causal effect is local or more global (i.e., long-range) in real-world applications. I wonder if the proposed method could estimate long-range causal effects accurately.

4. What is the difference/connection between the proposed method and targeted indirect experiment design in an active learning setting?

[1] Aglietti, V., Lu, X., Paleyes, A., & González, J. (2020, June). Causal bayesian optimization. In International Conference on Artificial Intelligence and Statistics (pp. 3155-3164). PMLR.

**Limitations:**

The authors adequately addressed the limitations of their work.

---

> ### Author Rebuttal · Authors · 2024-08-07
>
> We thank the reviewer for their kind assessment and useful comments. We reply to the raised questions one by one.
>
> **Causal Structure and Assumptions:** As long as the instrumental variable assumptions hold, we are agnostic to any confounding $U$ between the treatment variable $X$ and the outcome $Y$. We agree that the IV assumptions are rather strong and “valid instruments are hard to come by” (l.129, l.404-405). However, this issue is less severe in our setting than it is in other applications, as we control/design the instrument, e.g. the independence assumption $Z \perp U$ holds by simply randomizing the instrument. Moreover, we think it is reasonable that domain experts would not include instruments that may have no effect at all on $X$, e.g. a biologist may know the gene that a CRISPR guide is targeting, or the protein to which a compound binds. The exclusion restriction ($Z \perp Y \mid X, U$) is a potential constraint (e.g., l.125). However, as we consider multi-dimensional treatments we can make the exclusion restriction more realistic as we may add all known factors via which $Z$ may affect $Y$ into $X$. A useful example are orally administered antibiotics in so-called “sub-therapeutic dosages”: with the gut microbiome as $X$, it is reasonable to assume the antibiotics only affect macroscopic health outcomes via changes to the gut microbiome.
>
> **Sensitivity**: Sensitivity analyses are an important direction for further work. Frameworks (e.g., [1]) analyze the sensitivity of IV estimates to exclusion restriction violations, but are limited to linear settings and do not apply here. Extensions to non-linear systems (e.g., [2]) are limited and, in particular, cannot handle multi-dimensional treatments or underspecified settings. Developing such sensitivity analyses for point estimates and especially for bounds as in our setting would be a significant contribution.
> There is also an active area looking into other types of partial identification in the IV setting (e.g., [3], where partial identifiability stems from dropping the additive noise assumption). While additive (measurement) noise may actually be a defendable assumption in the settings we’re aiming at, major technical challenges remain in the multiply nested optimizations that are required to perform sensitivity analysis (in the case of [3], requiring a nested optimization in and of itself) on top of our adaptive procedure.
>
> **Scalability:**  We conducted higher-dimensional experiments as suggested, presenting results in the main rebuttal pdf. Fig. 1 and 2 show results for settings with $d_z = 5, d_x = 20$ and $d_z=20, d_x=20$, demonstrating the method's scalability beyond low-dimensional settings. In the $d_z=5, d_x=20$ setting, our adaptive approach effectively rules out the functional being negative or zero. The causal MSE is the smallest for our adaptive procedure on average, with the smallest variance across runs.
> We also report runtimes, highlighting that while runtime does increase in higher-dimensions, this increase is extremely mild. A key driver of the runtime is still the number of samples per experiment. As this may grow exceptionally large in certain applications, one may leverage existing techniques to handle the inversion of large kernel matrices, i.e. [4, 5]. Importantly, we note that computational cost is mostly a concern in synthetic experiments, as the time and cost to run physical experiments in real-world applications dwarf computational cost, which are at most polynomial in their input size. We will include these clarifications as well as the additional experimental results.
>
> **Causal Bayesian Optimization**: While there are clearly parallels between our work and Causal Bayesian Optimization, we believe that the setup (and thus the resulting methodology) are still distinct:
> (a) Their main goal is to maximize a target outcome, i.e., to set variables to certain levels that maximize the expected value of another variable $Y$ in the causal graph. Our goal is to maximally inform a causal mechanism (the function that determines $Y$) with no regard for what value $Y$ takes.
> (b) They assume feasible interventions on all graph variables. In contrast, we note that such interventions are often unrealistic, and experimentation amounts to “perturbations” (instrumenting) rather than “intervening”. If interventions on any variable were possible, we would directly intervene on $X$ for a trivial solution. Note that direct intervention allows them to consider general graphs, while our methodology pertains to perturbing/instrumenting a single treatment that directly affects the outcome.
> (c) In contrast to their approach, we do have a path dependence in our setting, i.e. our objective function does depend on the experiments we have performed so far. Due to this difference, our adaptive algorithm is closer to reinforcement learning instead of bayesian optimization or active learning where the objective function does not change depending on previous trials.
> We will add Causal Bayesian Learning to the related work section and clarify these differences.
>
> **global causal queries**: To identify global functionals, $h(z)$ with $z \sim \pi$ must put sufficient mass on all relevant treatment space regions for $Q$. If this region is large, more samples per experiment may be needed for reliable estimates. In practice, especially when $Z$ is much lower-dimensional than $X$, finding perturbations that cover the relevant $X$ regions can be difficult. In general, bounding nonlinear global functionals is possible, but still remains an important direction for future work, except for the linear case [6], where global properties can be inferred via linear extrapolation.
>
> **active learning**: See response to Causal Bayesian Optimization. One additional difference is the assumption of no unobserved confounding in active learning which we can relax via the use of instrumental variables. We will clarify these differences in the manuscript.

---

> ### Author Response · Authors · 2024-08-07
>
> [1] Cinelli, Carlos, and Chad Hazlett. "An omitted variable bias framework for sensitivity analysis of instrumental variables." Available at SSRN 4217915 (2022).
>
> [2] Vancak, Valentin, and Arvid Sjölander. "Sensitivity analysis of G‐estimators to invalid instrumental variables." Statistics in Medicine 42.23 (2023): 4257-4281.
>
> [3] Kilbertus, Niki, Matt J. Kusner, and Ricardo Silva. "A class of algorithms for general instrumental variable models." Advances in Neural Information Processing Systems 33 (2020)
>
> [4] Drineas, P., Mahoney, M.W. (2005). On the Nystrom Method for Approximating a Gram Matrix for Improved Kernel-Based Learning, 2005, Journal of Machine Learning Research, http://jmlr.org/papers/v6/drineas05a.html
>
> [5] Li, Mu et al. “Large-Scale Nyström Kernel Matrix Approximation Using Randomized SVD.” IEEE Transactions on Neural Networks and Learning Systems 26 (2015): 152-164.
>
> [6] Elisabeth Ailer, Jason Hartford, and Niki Kilbertus. 2023. Sequential underspecified instrument selection for cause-effect estimation. In Proceedings of the 40th International Conference on Machine Learning (ICML'23), Vol. 202. JMLR.org, Article 19, 408–420.

---

### Official Review · Reviewer_pD3c · 2024-07-12

**Soundness:** 3
**Presentation:** 3
**Contribution:** 3
**Rating:** 7
**Confidence:** 3

**Summary:**

The authors provide a procedure to use a sequence of encouragement designs to identify target functionals about a particular causal relationship.

**Strengths:**

This is a really cool problem setting. It isn't obvious to me that it's particularly common, but I think the larger idea of trying to think about _which_ experiments to run in order to gain knowledge about the world is a worthy line of inquiry.

In general, I think the approach of getting partial identification on a parameter and then determining a series of actions to take in order to reduce the width of that bound is a fantastic approach. This is a great way to conceptualize a variety of problems, I suspect.

**Weaknesses:**

It isn't clear to me why Equation 4 is a bound on Q[f_0]. Maybe this should be obvious to me, but I think if it isn't obvious to me, it probably won't be obvious to a lot of your readers, as I think I've read more into this literature than most people. This feels like your big contribution: after this bound is setup, the remainder isn't what I would call straightforward, but it feels more like standard machinery that I'm used to. Conditional on the bound, Theorem 1, 2 and the experiment selection procedures you provide all make a lot of sense. But I just don't see where this bound comes from. I would like to see (i) a clearer explanation of why this expression bounds Q[f_0], (ii) some clearer examples of the cases in which the bounds are equal and Q[f_0] is identified. I think it might also help to lay out when a non-optimal policy (what you call "non-informative experimentation") doesn't identify Q[f_0]. This is a bit confusing to me, as I'd expect that complete randomization as a policy would provide identification so long as there is positivity across the whole space: one could do something like off-policy evaluation to identify any particular \pi(Z). I believe the problem comes with the restriction discussed at the start of Section 3.2, but I do not see why it is that the solution to ensure that r_0 lies in TT^* is to take the +\- of Q[f] in the objective as from Eq 3. There's a connection here that you have not clearly spelled out to me.

Maybe its possible that the problem is that I need to read Bennett et al 2023 much more carefully to see why this follows, but that isn't a fair ask for your readers who are reading _this_ paper rather than that one.

Unfortunately, it's quite difficult for me to evaluate the overall novelty of this work without understanding this component. I think you have something very cool here, but I can't quite work that out based on the paper as it stands. I'd like to reiterate that I follow what you're doing at a high level (i.e. line 96-106 makes sense in general), but I don't follow when you get into specifics.

Some other miscellaneous thoughts that aren't as important:
- Do you _have_ to call this (\partial_i f)(x^*) a local effect? Between LATEs and everything else, this feels like an absurdly overloaded term.
- The experiments section makes it difficult to see clear quantitative comparisons between methods. I would like to see things like causal mean-squared error. I recognize that's a bit difficult when you're doing partial-ID, but demonstrating that the bounds collapse to the correct Q[f_0] is important. Even just doing something like showing the midpoint of the bounds quantitatively (with confidence interval based on replicates) and the width of the bound would be useful to make sure the process is behaving sensibly.

**Questions:**

see above

**Limitations:**

see above

---

> ### Author Rebuttal · Authors · 2024-08-07
>
> We thank the reviewer for their kind assessment and useful comments. We reply to the raised questions one by one.
>
> **Eq. 4 giving valid bounds**: Thanks for pointing out the missing steps here. Indeed, much of the machinery to see that eq. (4) yields valid bounds is in the Bennet et al. (2023) paper. Essentially, we follow the steps in their Section 3 replacing $\tfrac{1}{2} \langle h, h\rangle_{L_2}$ with $Q[h]$. (They use $h$ instead of $f$, apologies for that!) Then, the first equation in their Section 3 makes it more obvious that the constrained optimization yields bounds on $Q[f_0]$ as they read: “Minimize/maximize $Q[f]$ subject to $f$ satisfying the moment conditions, i.e., being compatible with the IV assumptions and data.” The realizability assumption then ensures that the optima of these optimizations are valid bounds on $Q[f_0]$. From here on, the solution to this problem is shown to be equivalent to the solution of our eq. (4).
> We follow Bennet et al. in (their) eq. (3), in using the method of Lagrange multipliers to obtain an alternative minimax formulation of the problem. Note that so far they do this in $L_2$, spaces. Analogous to their eq. (4), we can then replace the resulting estimator with a finite sample version and restrict ourselves to some function classes for $h$ (our $f$) and $g$. They then view the $L_2$ norm term as a penalization term (“so we call our estimator a penalized minimax estimator”) and we do the same for $Q[f]$. Up until this point, this distinction did not matter for the theory. As Bennet et al. note, the key distinction from previous work is to use the method of Lagrange multipliers. These steps demonstrate that – if we manage to show their eq. (5) for our modified Lagrangian – our minimax problem in (our) eq. (4) indeed yields bounds on $Q[f_0]$. In Section 4, Bennet et al. then go into showing that the problem can indeed be formulated as the minimax problem using the assumptions stated there (which we also had to carry over to our work). The results there (in particular Lemma 3) also directly apply to our modified Lagrangian. The key point in which their proof of the key identification Theorem 1 can fail in our modified setting is when they leverage the strong convexity of their Lagrangian in $h$ (our $f$), which comes from the $\langle h, h, \rangle_{L_2}$ term. Hence, as we discuss in “(In)validity of bounds” (specifically l.270 onwards), not every $Q$ will guarantee a unique optimum and valid bounds for eq. (4). First, valid bounds can be recovered for strictly convex functionals $Q$ (following the proof by Bennet et al. 2023). Second, we argue in the same paragraph why even for functionals that are not strictly convex, our method remains highly useful. We’re happy to discuss these arguments further.
>
> We agree that this line of reasoning is not self-contained in our paper and we will add a section in the appendix cleanly working out the computations for our Lagrangian explaining that we closely follow the work of Bennet et al. 2023.
>
> **non-informative policies and identification**: Thanks for the suggestion. We will add the following trivial examples in the appendix.
> 1) *Unidentified:* Consider $Z, Y \in \mathbb{R}$, $X \in \mathbb{R}^2$ and $h(z) = (z, 0)$, $f(x_1, x_2) = 0.5 * x_1 + 2 * x_2$ and $Q[f] = \partial_2 f (x^*)$. In this fully linear setting, due to the structure of $h$ the instrument can only ever perturb the 1st component of $X$ regardless of what policy we choose. Hence, it is impossible to identify the derivative of $f$ w.r.t. the second argument, which is just constant 2 (regardless of $x^*$). Even a policy with full support will not (even partially) identify $Q[f_0]$.
> 2) *Fully identifiable:* Consider a similar setting as above, but with $f(x_1, x_2) =  x_1^2 + x_2$ and $Q[f] = \partial_1 f(x^*)$. Let’s say $x^* = (6, 1)$, then $Q[f_0] = 2 \cdot 6 = 12$. Any policy $\pi$ that has positive density in a neighborhood of $z=6$ will be able to fully identify $Q[f_0]$. However, any policy that puts no mass (or in practice, little mass) near $z=6$, will not identify $Q[f_0]$. This would be an uninformative policy.
>
> **”local” effect**: Indeed, this is not the best wording in this context. We’ll think of a more distinct alternative (not so easy actually).
>
> **experiments and metrics**: Thanks for this suggestion. We included the causal mean-squared error in the plots in the main rebuttal pdf as well as in the revised version of the paper. As expected, our adaptive method not only achieves the best average causal MSE, but also exhibits the least variance across runs. We are still working on – and plan to also include – the suggested plots with the midpoint between bounds + ranges as intervals to present results more clearly.
>
>
> [1] Luenberger, David G. Optimization by vector space methods. John Wiley & Sons, 1997.
>
> [2] Elisabeth Ailer, Jason Hartford, and Niki Kilbertus. 2023. Sequential underspecified instrument selection for cause-effect estimation. In Proceedings of the 40th International Conference on Machine Learning (ICML'23), Vol. 202. JMLR.org, Article 19, 408–420.

---

> > ### Comment · Reviewer_pD3c · 2024-08-11
> >
> > I thank the authors for their clarifications, I think I see more clearly where the bound is coming from. I'll raise my score given this.
> >
> > To put it a bit more plain (to me), you're taking the minimum and maximum norm solutions (by changing the sign of Q[f] in Eq.4) from within the satisfying set of functions $f$ and $g$. Under full identification, those two would optimands coincide, as there is only one satisfying function, but under partial identification that may not be the case.
> >
> > I also think the comment the authors provided to reviewer GhAo regarding the CRISPR use-case is interesting. Coming from more of a social science perspective, this is not a setting which would come up much, but I suspect this may be different in the case you're considering. Making this more clear in the text why this setting matters would help the paper's impact.

---

> ### Author Response · Authors · 2024-08-12
>
> We thank the reviewer for their comment and increasing their score. The intuition about the bounds is very well put. We highly appreciate the suggestion and will include the description of the CRISPR use-case into the revised version.

---

### Official Review · Reviewer_GhAo · 2024-07-30

**Soundness:** 4
**Presentation:** 3
**Contribution:** 3
**Rating:** 7
**Confidence:** 3

**Summary:**

The authors' primary goal is to design experiments that maximally inform a query of interest about the underlying causal mechanism, within a fixed experimentation budget. They address this by maintaining upper and lower bounds on the query and sequentially selecting experiments to minimize the gap between these bounds. They show that by treating experiments as instrumental variables, they can estimate these bounds using existing techniques in nonlinear instrumental variable estimation. Their procedure involves a bi-level optimization: an inner optimization estimates the bounds, while an outer optimization seeks to minimize the gap between them. For certain queries, when assuming the underlying function lies within a reproducing kernel Hilbert space (RKHS), they derive closed-form solutions for the inner estimation problem. The authors develop adaptive strategies for the outer optimization to iteratively tighten these bounds and demonstrate empirically that their method robustly selects informative experiments, leading to identification of the query of interest when possible within the allowed experimentation framework.

**Strengths:**

S1. The presentation is generally clear, with a logical structure that guides readers through the problem formulation, methodology, and initial experimental results.

S2. The authors present a novel approach to causal effect estimation in nonlinear systems with potential unobserved confounding. This framework addresses some challenges in scientific settings where direct experimentation is not possible, potentially aiding more targeted scientific discovery.

**Weaknesses:**

W1. The paper only demonstrates results on a low-dimensional synthetic setting (d_x = d_z = 2). While this provides a proof-of-concept, it's unclear how the method scales to higher dimensions or performs on real-world data.
The authors could:
- Extend experiments to higher dimensions (e.g. d_x, d_z = 10 or 20) to show scalability.
- Test on semi-synthetic data by using real covariates with simulated outcomes.
- Discuss computational complexity as dimensionality increases.

W2. The paper makes strong assumptions (e.g. RKHS, valid IVs) without thoroughly discussing their implications. The authors could:
- Provide sensitivity analyses for key assumptions.
- Discuss scenarios where assumptions may not hold in practice.
- Clarify which parts of the method rely on which assumptions.

**Questions:**

No.

**Limitations:**

No.

---

> ### Author Rebuttal · Authors · 2024-08-07
>
> We thank the reviewer for their kind assessment and useful comments. We reply to the raised questions one by one.
>
> **Scalability:** Following the suggestion, we have worked on additional higher-dimensional experiments. We show all results in the main pdf of the rebuttal. In Fig. 1 and 2, we present results for higher-dimensional settings with $d_z = 5, d_x = 20$ as well as $d_z=20, d_x=20$. Overall, these demonstrate that the method can indeed scale beyond the proof-of-concept low-dimensional setting. Specifically, in the $d_z=5, d_x=20$ setting, our adaptive approach is the only one that essentially rules out the functional being negative (or zero). Similarly, the causal MSE is not only the smallest for our adaptive procedure on average but also has the smallest variance across runs.
> Finally, we report runtimes for these settings as well, highlighting that while runtime does increase in higher-dimensions, the increase is extremely mild. A key driver of the practical runtime is still the number of samples per experiment. As this may grow exceptionally large in certain applications, one may leverage existing techniques to efficiently handle the inversion of large kernel matrices, i.e. [1, 2] to speed up computations. Importantly, we note that computational cost is mostly a concern in our synthetic experiments, as we imagine the time and cost to run physical experiments in real-world applications dwarf computational cost and time, which are at most polynomial in their input size (like kernel methods). We will include these clarifications as well as the additional experimental results (in polished figures) to the final paper.
>
> **Strong Assumptions:** We agree that obtaining strong guarantees in causality typically requires strong assumptions and that our methodology may be restricted by those (as mentioned in l.125, 129). We have the following comments:
>
> 1) *valid IVs*: Indeed “valid instruments are hard to come by in practice” (l.129, see also l.404-405). At the same time, we believe this problem to be less grave in our setting than it is in other applications, where one hopes to “accidentally find valid instruments naturally”. In our setting, we control/design the instrument. Therefore, the independence assumption $Z \perp U$ actually holds (by simply randomizing the instrument). Similarly, it is reasonable that domain experts would not include instruments in the considered domain that may have no effect at all on $X$, i.e., we believe it is not outrageous to defer to practitioners to only consider fairly strong instruments for which $Z \not \perp X$ holds. For example, a biologist may know the gene that a CRISPR guide is targeting, or the protein to which a compound binds. Off-target effects remain a very real concern, but when IV’s are experimentally selected, we can hope to select IVs with the IV assumptions in mind to minimize  exclusion violations.
> As mentioned in the paper (e.g., l.125), the exclusion restriction ($X \perp Y \mid X, U$) remains a potentially restricting factor. At the same time, the fact that we may consider multi-dimensional treatments can effectively make the exclusion restriction more realistic as we may add all known factors via which $Z$ may affect $Y$ into $X$. A useful example are orally administered antibiotics in so called “sub-therapeutic dosages”. This means that the antibiotics enter the stomach and gut, but cannot be detected in the blood stream. When using the gut microbiome for $X$, it is then reasonable to assume that the antibiotics only affect macroscopic health outcomes via the changes to the gut microbiome.
>
> 2) *Sensitivity*: We agree that sensitivity analyses towards these assumptions are an important direction for further work. In terms of the exclusion restriction, there has been a line of works (recently summarized and extended into a usable framework by [4]) providing tools to analyze the sensitivity of IV estimates to violations of the exclusion restriction. However, these works are limited to the linear setting and thus do not apply to our setting. Rare extensions to non-linear systems (e.g., [5]) only apply to limited settings and in particular cannot handle multi-dimensional treatments or underspecified settings. Providing such sensitivity analyses not only for point estimates, but to the bounds in our underspecified setting with multi-dimensional treatments would likely be a substantial contribution in and of itself.
> There is also an active area looking into other types of partial identification in the IV setting (e.g., [6], where partial identifiability stems from dropping the additive noise assumption). While additive (measurement) noise may actually be a defendable assumption in the settings we’re aiming at, major technical challenges remain in the multiply nested optimizations that would be required to perform sensitivity analysis (in the case of [6], requiring a nested optimization in and of itself) on top of our adaptive procedure.
>
> 3) *Source condition*: The practical limitations of the source condition are not conclusively resolved or at least still debated. To the best of our knowledge, [3] are the first to relax this assumption. Some of the ideas there may transfer to our setting to relax the source condition, but we consider this a direction for future work that likely requires substantial novel theoretical developments.
>
> 4) *RKHS*: We do not fully understand the restrictions the reviewer is pointing towards by assuming RKHSs. Broadly speaking, RKHS are in our experience generally viewed as a reasonable and flexible assumption for many practical purposes. For characteristic/universal kernels (like the RBF we use), they enjoy universal approximation properties and often have strong consistency and convergence guarantees (see [8] for an overview). Together with the empirical success of kernel-based methods in many domains and the efficient existing algorithms for practical estimation, we believe them to be a reasonable choice.

---

> > ### Comment · Reviewer_GhAo · 2024-08-13
> >
> > Thank you for your detailed explanation, which has addressed my concerns. I ask this question because recent research in social sciences has highlighted that the validity of instrumental variables remains a significant challenge [1]. I have increased my score accordingly.
> >
> > Also, CRISPR example piqued my interest. I'm curious about how your work relates to or differs from research in biostatistics centered on Mendelian randomization (if any).
> >
> > [1] Rain, rain, go away: 194 potential exclusion-restriction violations for studies using weather as an instrumental variable. Jonathan Mellon.

---

> ### Author Response · Authors · 2024-08-07
>
> [1] Drineas, P., Mahoney, M.W. (2005). On the Nystrom Method for Approximating a Gram Matrix for Improved Kernel-Based Learning, 2005, Journal of Machine Learning Research, http://jmlr.org/papers/v6/drineas05a.html
>
> [2] Li, Mu et al. “Large-Scale Nyström Kernel Matrix Approximation Using Randomized SVD.” IEEE Transactions on Neural Networks and Learning Systems 26 (2015): 152-164.
>
> [3] Bennett, Andrew, et al. "Minimax Instrumental Variable Regression and $ L_2 $ Convergence Guarantees without Identification or Closedness." The Thirty Sixth Annual Conference on Learning Theory. PMLR, 2023.
>
> [4] Cinelli, Carlos, and Chad Hazlett. "An omitted variable bias framework for sensitivity analysis of instrumental variables." Available at SSRN 4217915 (2022).
>
> [5] Vancak, Valentin, and Arvid Sjölander. "Sensitivity analysis of G‐estimators to invalid instrumental variables." Statistics in Medicine 42.23 (2023): 4257-4281.
>
> [6] Kilbertus, Niki, Matt J. Kusner, and Ricardo Silva. "A class of algorithms for general instrumental variable models." Advances in Neural Information Processing Systems 33 (2020)
>
> [7] Sriperumbudur, Bharath K., Kenji Fukumizu, and Gert RG Lanckriet. "Universality, Characteristic Kernels and RKHS Embedding of Measures." Journal of Machine Learning Research 12.7 (2011).
>
> [8] Schölkopf, Bernhard, and Alexander J. Smola. Learning with kernels: support vector machines, regularization, optimization, and beyond. MIT press, 2002.

---

> ### Author Response · Authors · 2024-08-14
>
> We thank the reviewer for the comment and increasing their score.
> The connection to Mendelian Randomization can be drawn via the Instrumental Variable setup we use as experiments. MR is a special case of instrumental variables in which genetic variants are used as instruments. Therefore---in theory---our method would propose the next genetic variant which would inform the estimator the most. In practice, MR relies on existing genetic variation within a population and not so much on designing genetic variation. The difference therefore lies in the assumption of the instrument: in our setting we are able to adjust the instrument/experiment, MR looks at existing genetic variation. We will add this aspect in the manuscript as both approaches target similar questions. Thank you for the comment.

---

### Author Rebuttal · Authors · 2024-08-07

We thank the reviewers for their valuable feedback. We included the pdf showcasing various additional experiments in response to specific questions and kindly refer to the individual responses. Due to the space restrictions for the individual rebuttals, we had to heavily compress our replies to the point where we could not express everything we planned to comment on and important details or nuance may have gotten lost. We are more than happy to elaborate on any of the answers in more detail.

---

### Decision · Program_Chairs · 2024-09-25

**Decision:**

Accept (poster)

**Comment:**

All reviewers agree that this is a solid paper with novel and interesting contributions. There are certainly some weaknesses, such as the limited experimental validation involving only  synthetic and low-dimensional settings, or questions about the validity of some assumptions. On the other hand, the authors could address at least some of these critical comments in a rather convincing way, so that for me as an AC, in the end the positive aspects clearly outweigh the points of criticism. Therefore I recommend acceptance of this paper.